# C³ONTEXT: A Common Consensus on Convective OrgaNizaTion during the EUREC⁴A eXperimenT

Hauke Schulz[1]

[1]Max Planck Institute for Meteorology, Hamburg, Germany

**Correspondence:** Hauke Schulz (hauke.schulz@mpimet.mpg.de)

**Abstract.** The C³ONTEXT (A Common Consensus on Convective OrgaNizaTion during the EUREC⁴A eXperimenT) dataset is presented as an overview about the meso-scale cloud patterns identified during the EUREC⁴A field campaign in early 2020. Based on infrared and visible satellite images, 50 researchers of the EUREC⁴A science team manually identified the four prevailing meso-scale patterns of shallow convection observed by Stevens et al. (2020). The common consensus on the observed meso-scale cloud patterns emerging from these manual classifications is presented. It builds the basis for future studies and reduces the subjective nature of these visually defined cloud patterns. This consensus makes it possible to contextualize the measurements of the EUREC⁴A field campaign and interpret them in the meso-scale setting. Commonly used approaches to capture the meso-scale patterns are computed for comparison and show good agreement with the manual classifications. All four patterns as classified by Stevens et al. (2020) were present in January - February 2020 although not all were dominant during the observing period of EUREC⁴A. Supplemental classifications of storm-resolving simulations suggest that the latter have limited ability to replicate the observed cloud patterns and require further research.

The full dataset including postprocessed datasets for easier usage are openly available at the Zenodo archive at https://doi.org/10.5281/zenodo.5979718 (Schulz, 2022a).

## 1 Introduction

Clouds are often clustered. Examples of larger clusters are squall lines and the intertropical convergence zone. But also much smaller clouds like shallow trade wind cumuli, are often seen clustered on a scale of several hundred kilometers. The understanding of these meso-scale patterns of shallow convection is still sparse. However, the ubiquity of these clouds and their reoccurring structure suggest that they play an important role in determining the radiative effects of the trade-wind regimes (Bony et al., 2020). The EUREC⁴A (ElUcidating the RolE of Cloud-Circulation Coupling in ClimAte) field campaign addresses among others, the question of which processes are at play that change the meso-scale appearance of shallow convection. Prior to the campaign, studies concentrated on the classification of meso-scale patterns based on satellite images by visual inspection (Stevens et al., 2020), rule-based algorithms (Bony et al., 2020; Janssens et al., 2021), supervised neural networks (Rasp et al., 2020) and unsupervised neural networks (Denby, 2020). Most of these methods focus on four specific meso-scale cloud patterns that have been identified as commonly occurring in the North Atlantic downwind trades (Stevens et al., 2020). An

overview about these four patterns which are named by their visual impressions as *Sugar*, *Gravel*, *Flowers* and *Fish* is given
    in Fig. 1.

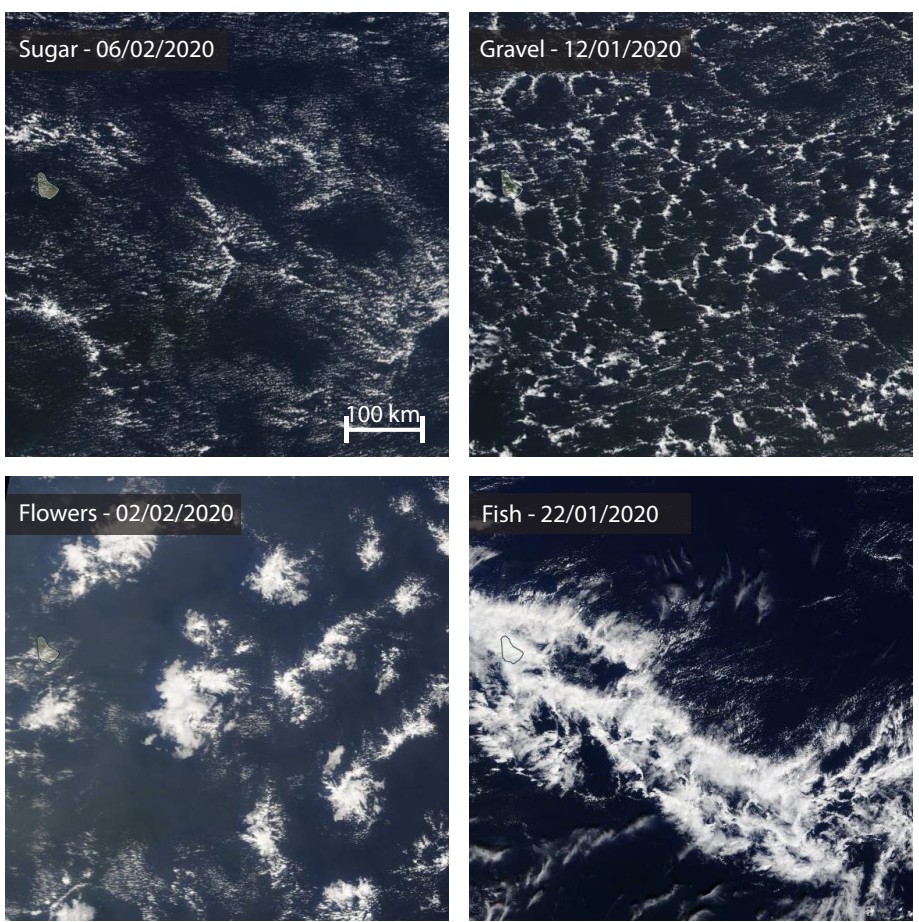

**Figure 1.** Four classes of meso-scale patterns of shallow convection have been labeled by 50 scientists in satellite images of January and
February 2020. An example of each of these classes is shown based on the reflectance measurements of the Moderate Resolution Imaging
Spectroradiometer (MODIS) onboard the Terra satellite (Sugar, Gravel) and the Visible Infrared Imaging Radiometer Suite (VIIRS) onboard
the Suomi NPP satellite (Flowers, Fish). Example images provided by NASA Worldview.

These efforts show that the categorization of the meso-scale patterns is an elementary step in acquiring further knowl-
edge about the cloud processes in the downstream trades. 50 scientists from 15 research institutes who were involved in the
EUREC[4]A field campaign in January - February 2020 therefore participated in a joint online classification event. The re-
sults of this classification event are presented here. Due to the high attendance, the presented dataset can be interpreted as
the common judgement of the EUREC[4]A science team on the meso-scale patterns of shallow convection in the wider trades
during EUREC[4]A. The studied domain of 5°N - 20°N and 62°W - 40°W contains the entire research area, including both
the *Tradewind alley* in the north (12.5°N - 14.5°N) and the *Boulevard des Tourbillions* close to the coast of South America

(Stevens et al., 2021a). This dataset therefore allows to consistently communicate about the otherwise subjectively defined patterns and puts the measurements taken during EUREC⁴A in their meso-scale context to advance the process understanding of shallow convection in the trades. The need for a better process understanding is highlighted by a supplemental classification of storm-resolving simulations covering the same time-period.

The paper is structured as follows: the dataset and its collection are described in Section 2. Potential use cases are described in Section 3. In Section 4 additional classification methods that are able to detect the four meso-scale patterns are applied to the EUREC⁴A time period and compared to the manual classifications described in this paper. We conclude with Section 5.

## 2 Data description and development

The manual classifications were gathered through the online platform zooniverse.org which has already been successfully used in an earlier project by Rasp et al. (2020). The platform makes it possible to crowd-source labels for e.g. machine learning projects. Additional workflows can be defined to separate different image sources or to separate, for example, labels made during a practice run from those that belong to the actual classification. The former allowed everyone to familiarize with the zooniverse.org platform without influencing the results.

For this dataset, we defined three workflows. Two workflows are based on satellite observations. The first one is based on the visible channels of both the Geostationary Operational Environmental Satellite 16 (GOES-16) Advanced Baseline Imager (ABI) and Moderate Resolution Imaging Spectroradiometer (MODIS) onboard the Aqua and Terra satellites and is called *EUREC⁴A VIS*. The second one is based on the "clean" infrared channel (channel 13) of GOES-16 ABI only (*EUREC⁴A IR*). To test how well storm-resolving simulations are able to reproduce the observed patterns, we included another workflow, *EUREC⁴A ICON*, which is based on ICON storm-resolving simulations covering the EUREC⁴A time period. The simulation has a grid-spacing of $1.25\,\mathrm{km}$ and was initialized daily by the ECMWF IFS and forced hourly. Each run is 48 hours long. The first 24 hours were discarded to allow for spin-up.

To visualize the simulation output, we calculated a pseudo-albedo $\alpha$ by following the approximation of Zhang et al. (2005):

$$\tau = 0.19 \cdot \mathrm{LWP}^{\frac{5}{6}} \cdot N^{\frac{1}{3}} \tag{1}$$

$$\alpha = \frac{\tau}{6.8 + \tau}, \tag{2}$$

where $\tau$ is the optical depth, LWP is the liquid water vapor path and $N$ an assumed cloud droplet number density. Here a number density of $70\,\mathrm{cm}^{-3}$ is used, which is at the lower end of the observed concentrations (Siebert et al., 2013).

All workflows are further described in Tab. 1.

On March 24, 2020 the international, virtual classification event was hosted with 50 scientists from 15 institutes participating to create the pattern classifications. For a full day the participants classified patterns of shallow trade-wind convection by labeling, i.e. drawing rectangles of variable sizes around the four common types: *Sugar*, *Gravel*, *Flowers* and *Fish* (Stevens et al., 2020).

**Table 1.** Description of data sources used to create the images of the classification workflows.

| | EUREC$^4$A VIS | | EUREC$^4$A IR | EUREC$^4$A ICON |
|---|---|---|---|---|
| | MODIS (Terra/Aqua) | ABI (GOES-16) | ABI (GOES-16) | |
| Domain | 5° N - 20° N; 62° W - 40° W | | | 5° N - 20° N; 62° W - 44° W |
| Period | 07.01.2020 - 22.02.2020 | | | 16.01.2020 - 20.02.2020 |
| Resolution | ~1 km | | | ~2 km |
| (shown) | 2-hourly, 12-20 UTC | 2-hourly | | 2-hourly |
| Data source | Corrected reflectance | Channel 02 (red) | Channel 13 (IR) | Pseudo-albedo |
| Number of scenes | 94 | 234 | 562 | 425 |
| Remarks | | | | Classified after 24 hours spin-up |

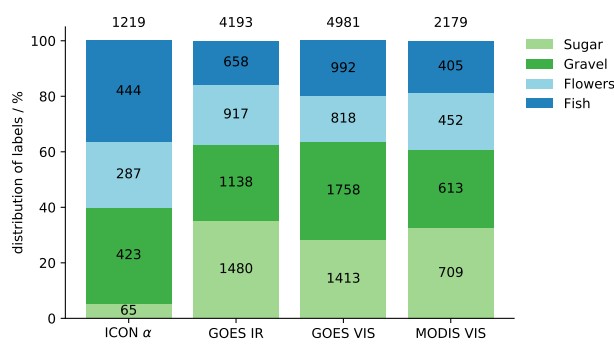

**Figure 2.** Distribution of labels by data source. The relative distribution is shown at the y-axis, while the absolute number of labels is indicated within each bar. The total number of labels per workflow is shown at the top of each bar.

In the end, over 12,500 labels were gathered. The observational workflows got the majority of the labels (see Fig. 2) because the identification of the patterns in the model simulation was too demanding. The cloud features in the simulations had too little similarity with those found in nature. Comparing the distribution of labels in the individual workflows, Figure 2 reveals that *Sugar* was disproportionately rarely identified in the ICON simulations. The largest feature, like *Fish*, however, has been identified more often. This supports the assumption that larger features are better reproduced in storm-resolving simulations than features of smaller scales, like *Sugar*.

Because all users have familiarity with the patterns either by previous work and/or being involved in the classification event of Rasp et al. (2020) it can be assumed that the labels are of high quality. In addition, they were trained immediately before starting the classification through an online presentation to get familiar with the labeling interface on zooniverse.org and to

refresh once more the different meso-scale cloud pattern categories. Compared to Rasp et al. (2020) where the focus has been to classify as many diverse cloud scenes as possible to capture nature's variability and thus serve as a better machine learning dataset, the aim for this dataset is to create a common classification dataset for the EUREC$^4$A time period that participating scientists agree on and can directly be used in further studies. Therefore, the temporal frequency has been increased from daily cloud scenes to 2-hourly cloud scenes to reflect also the changes on the sub-daily scale such as those identified by Vial et al.

(2021). Due to this design difference, a single day is now on average classified 20 times in case of the visible workflow instead of just about 6 (3 per each daytime Aqua/Terra satellite overpass) as in Rasp et al. (2020). Each individual image is however still viewed about three times.

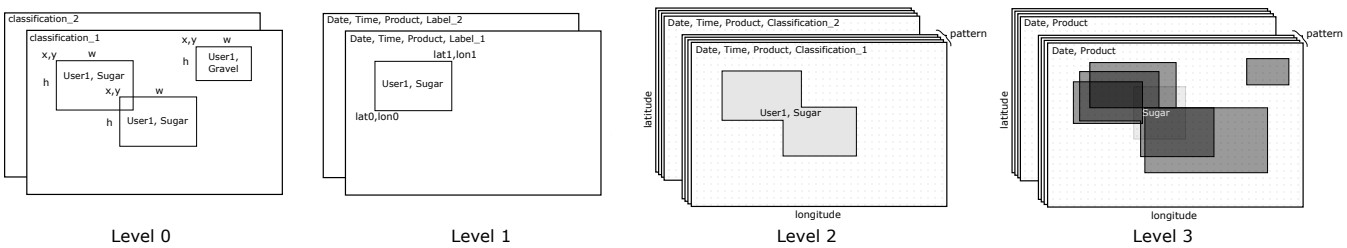

**Figure 3.** Overview of processing levels of the datasets including the variable names used in the respective datasets.

After the joint classification event, over 12,500 labels were processed to make them more user friendly, especially because

the raw data misses any temporal and geospatial information. The processing steps with the intermediate products are illustrated in Fig. 3 and described as follows:

- *Level 0*

  The *Level 0* dataset consists of the raw data output and originates from the zooniverse platform. It consists of CSV files that contain entries for each workflow, image (subject) and classification including technical details like the time spent

on drawing a specific label. Labels are given by their origin (`x,y`) and their height (`h`) and width (`w`) given in pixel coordinates.

- *Level 1*

  The *Level 1* dataset is further processed and combines the information distributed over the *Level 0* dataset files. It contains each label as a separate entry and contains information about the classified object, the user and the geographical

and Cartesian coordinates of the label. This product is saved in a netCDF file.

- *Level 2*

  For the *Level 2* dataset, the data are merged by `classification_id`.

  The `classification_id` is a unique identifier of a classification, where a classification refers here to the process of labeling a single image by a single user. The user might use several labels of the same or different kind to completely

classify a scene. This process eliminates overlaps of same-user classifications for each pattern and turns the data into masks, rather than coordinates (see Fig. 3). In cases where same-user classifications of different patterns overlap, the overlapping region is counted towards all classified patterns. This case is not handled specifically as it shows the uncertainty a user had to classify a specific region as one or the other pattern. Masks have the advantage to be easier queried whether a specific location is influenced by a meso-scale pattern or not. This product is saved as zarr.

    – *Level 3*

    To ease working with the dataset, the percentage of agreement ($p$) among users on a specific pattern on each location is calculated and saved as `Level 3` data for each workflow. It is calculated as follows:

$$p_{\text{pattern}}(i,j) = \frac{\sum_0^U c(i,j)}{U}, \tag{3}$$

    where $U$ is the number of users that have seen the particular image, $c$ the classification mask from the *Level 2* data and $i, j$ the geographic coordinates. Because the labels of users that attributed several classes to one pixel are not removed, $\sum p$ can be greater than 100%. An additional *Unclassified* category is introduced to capture the agreement on unclassified regions which is implicitly visible in e.g. Fig. 4. Fig. 4 shows how participants agreed on the different patterns on February 12 depending on the workflow. This figure is continued in Appendix A, where this overview of daily averaged classifications is given for each day to give an impression of the dataset and in particular the presence and distribution of meso-scale patterns during the EUREC4A field campaign. This product is saved as zarr.

## 3 Potential dataset use and reuse

The EUREC[4]A field campaign has been an international study with a wide range of research platforms and many minor objectives (Stevens et al., 2021b). This dataset does not only cover the core area of the experiment, but also the wider area and time period. While the participating research airplanes and drones were mostly staying in the trade-winds, some research vessels conducted measurements as far south as 6.5°N.

    This dataset gives the opportunity to study all these measurements in the context of the meso-scale patterns observed in the downwind trades. Due to the high subjectivity of these meso-scale cloud pattern definitions, it is of particular importance to discuss results based on a common consensus to keep studies comparable. The C[3]ONTEXT dataset can serve as such a reference for the period of the EUREC[4]A field campaign.

    Fig. 5 shows the meso-scale context for three platforms participating in EUREC[4]A and representing different measurement strategies (see Stevens et al. (2021a) for a complete list of platforms). Based on the classifications of the infrared workflow, the daily changes in the cloud patterns become apparent. The infrared workflow is preferred over the visible one, because it covers both day and night and is therefore able to also capture the dial nature of the patterns (Vial et al., 2021). Before the intense observation period (IOP), January 20th to February 20th, the prevailing cloud pattern at the Barbados Cloud Observatory was *Gravel*. With the start of the IOP, *Fish* was detected most in the research area. After January 24, patterns were much

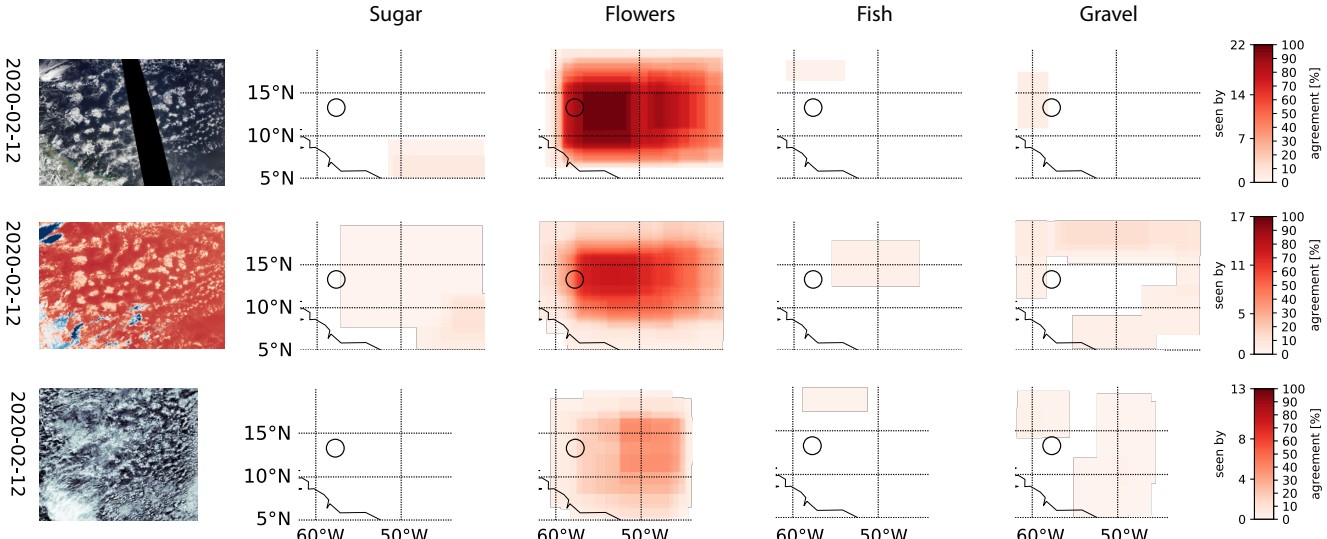

**Figure 4.** Manual classification examples for the three workflows (top to bottom: visible, infrared, simulation). The labels for each pattern (from left to right: *Sugar*, *Flowers*, *Fish*, *Gravel*) are shown next to the labeled image. Coastline of South America is marked in black. The circle marks the EUREC[4]A circle, one focus area of the EUREC[4]A field campaign and the main flight pattern of the participating research aircraft HALO (Konow et al., 2021). Coastlines are based on GSHHG shapefiles (Wessel and Smith, 1996).

less widespread and well defined, such that there was more disagreement on the patterns. This transition is independent of the location. Both the platforms in the north as well as the R/V Atalante, which sailed further south in the *Boulevard de Tourbillons (eddy boulevard)* which is defined by Stevens et al. (2021b) as the area of the coastline of Brazil where the North Brazil Current (NBC) Rings occur, were embedded in a similar transition. The reduced frequency of classifications towards the end of the IOP around February 15th are caused by mid-level clouds obscuring the view of shallow convection.

In Appendix A the *Level 3* products are visualized for each studied day and can be used as a look-up table to quickly identify if a pattern has been identified at a specific time and place.

## 4 Comparison with other classifications

Several methods have been developed to describe the meso-scale structure of shallow convection (Wood and Hartmann (2006); Rasp et al. (2020); Bony et al. (2020); Denby (2020)). Here we focus on the comparison of the manual classifications with two other methods that specifically aim to detect the four meso-scale patterns of the downwind trades as defined by Stevens et al. (2020) and shown in Fig. 1. Bony et al. (2020) combined a measure of organization ($I_{org}$, Tompkins and Semie (2017)) with the mean cluster size (S), while Rasp et al. (2020) developed a deep neural network to detect the patterns. Although Janssens et al. (2021) show that different combinations of metrics, like cloud fraction and fractal dimension, better describe the variance in a cloud field, the pair of $I_{org}$ and mean cluster size has been widely used and is considered here for better comparison.

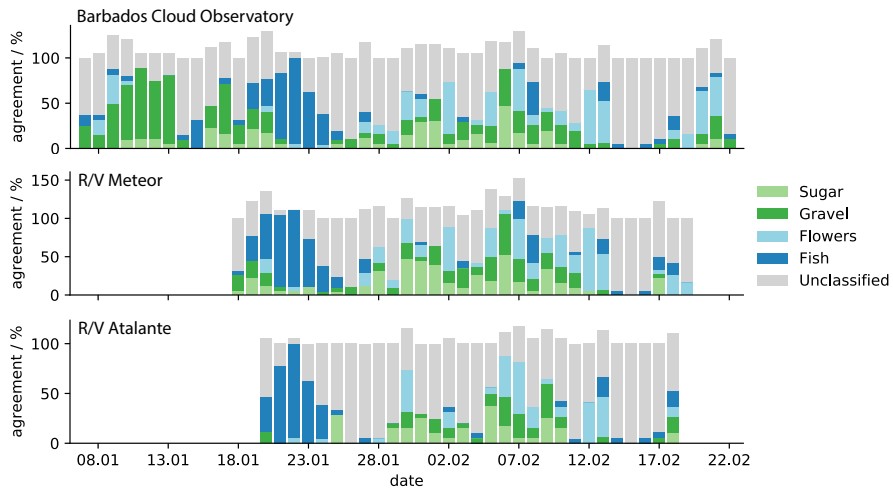

**Figure 5.** Exemplary use case of *Level 3* data: meso-scale setting of research platforms during EUREC[4]A based on the *EUREC[4]A-IR* workflow (top to bottom: Barbados Cloud Observatory (BCO), R/V Meteor, R/V Atalante)

To assess the agreement between the different classification methods we compare the method of Bony et al. (2020) and a deep neural network based on Rasp et al. (2020) that detects the patterns in geostationary infrared images of GOES-16 ABI (Schulz et al., 2021) instead of visible images taken onboard the polar-orbiting satellites Terra and Aqua. It should be noted that this deep neural network has not been trained with the manual classifications presented here, but with the manual classifications
of Rasp et al. (2020) which cover older years and included different geographical regions. The network is identical to the one used in Schulz et al. (2021).

Because the $I_{org}/S$ measure is sensitive to the domain size, we compute these metrics over a $10 \times 10$ degree sub-domain and consider only classifications within this domain for the comparison. Specifically, we focus on the region 10°N - 20°N and 58°W - 48°W. This domain size ensures that $I_{org}$ is describing the organization on the meso-scale and the cloud patterns we are
155 interested in. The method has been successfully used in this domain in Bony et al. (2020) and applied accordingly. Brightness temperatures between $280\,\mathrm{K}$ and $290\,\mathrm{K}$ are regarded as clouds. Days where the 25th percentile of brightness temperatures within any satellite image is lower than $285\,\mathrm{K}$ are discarded to avoid a bias by high clouds.

Fig. 6 shows the comparison of the different methods. From the appearance of the patterns, we expect *Gravel* and *Flowers* to be rather regularly distributed and therefore to have a lower $I_{org}$ compared to *Fish* and *Sugar*. It should be noted that $I_{org}$ is
160 calculated based on a threshold in brightness temperature and therefore only the deeper clouds in the *Sugar* field are considered leading to a higher $I_{org}$ than one would expect from an otherwise rather randomly distributed cloud field. The mean cluster size should be small for *Sugar* and *Gravel* and larger for *Flowers* and *Fish*.

Indeed, the pattern area fraction maximizes for each pattern in the respective quadrant of the $I_{org}/S$ space independent of the classification method. *Gravel* classifications dominate the lower left quadrant, *Sugar* dominates the upper left quadrant

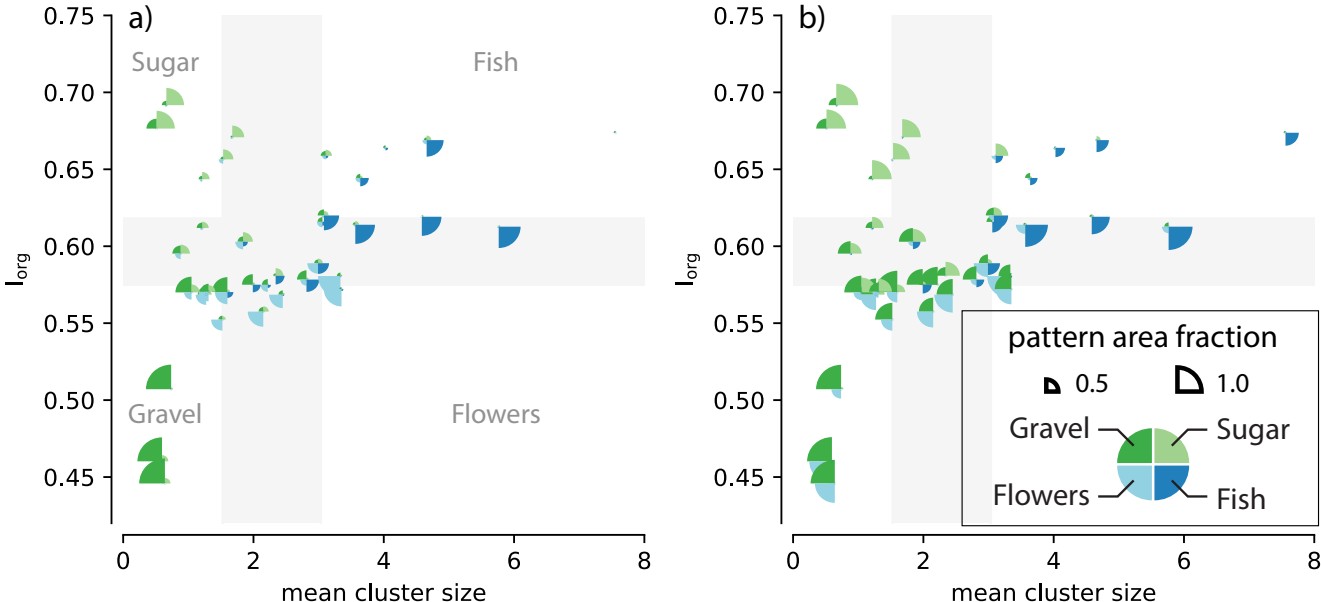

**Figure 6.** Comparison of the $I_{org}$ *versus mean cluster size (S)* classification method and the (a) *manually* classified patterns and (b) the neural network classifications. For each $I_{org}$/S classification, the manually/automatically classified pattern area for each pattern is indicated as wedges of different sizes. Mean cluster size is given as fraction of the domain size and multiplied by $10^4$ to be consistent with Bony et al. (2020). Grey areas indicate the middle terciles of $I_{org}$ and S and are the bounds of the four pattern quadrants. All classifications are based on the GOES-16 ABI infrared images and are daily averages.

and *Fish* the upper right one. *Flowers* are harder to associate with a quadrant as they are more centered. In general, the $I_{org}$/S distribution has a large range of values for small mean cluster sizes and narrows with increasing mean cluster size.

This is also in alignment with Bony et al. (2020) where the lower right quadrant includes not only *Flowers* but also about 35% of the *Fish* cases (their Fig. 1c). *Flowers*' cloud entities are generally smaller than those of *Fish* pattern and therefore cannot separate very well from the other patterns.

Fig. 7 reveals the time series of the area fraction covered by the meso-scale patterns within the region of 10°N - 20°N and 58°W - 48°W and identified by the different classification approaches. For a better comparison with the EUREC⁴A VIS workflow, Fig. 7 is only based on daytime classifications (12 - 20 UTC).

To convert the agreement on patterns in the *Level 3* dataset into actual classifications, we applied a threshold of 0.1 to the percent agreement, so that at least 10% of participants who viewed a given scene must have agreed on a pattern. Agreement

below this threshold is treated as unclassified region. This threshold should be seen as a lower bound and is only used to reduce the noise of the classifications. For comparison, we also apply a stricter threshold of 0.5, which removes most of the overlap (Fig. 7b). The overall day-to-day changes remain robust. Fig. 7 shows that in January - February 2020 all patterns

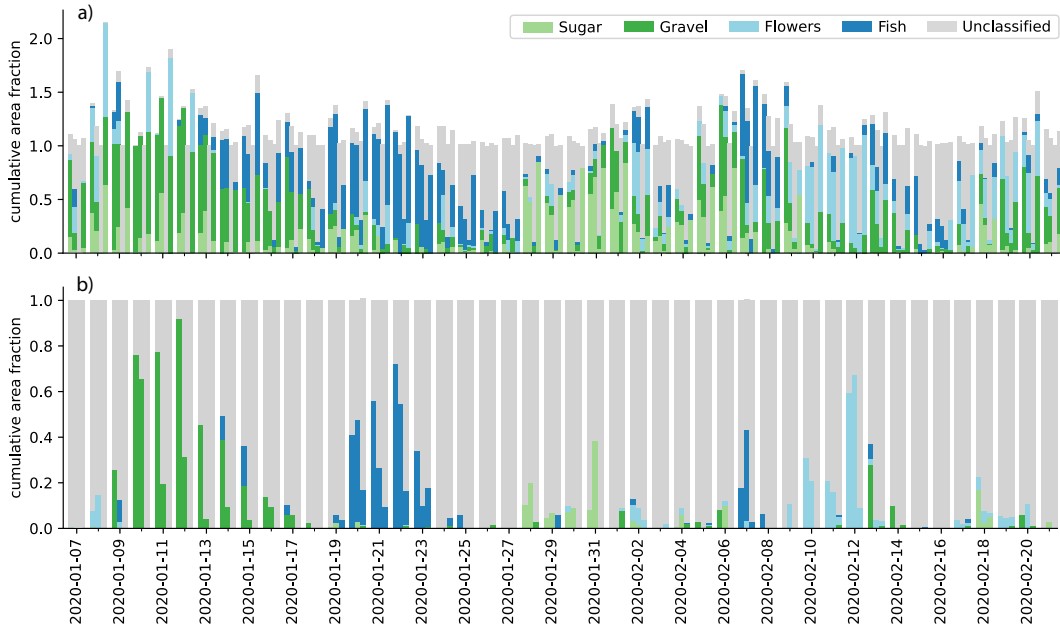

**Figure 7.** Time series of area fraction covered by each pattern as identified in the workflows EUREC[4]A VIS (first bar), EUREC[4]A IR (second bar), EUREC[4]A ICON (third bar) and the deep neural network trained for the infrared classification of cloud patterns (forth bar in a)) within the area of 10°N - 20°N and 58°W - 48°W. Thresholds of 0.1 (a) and 0.5 (b) have been applied to the percentage of agreement of the manual classifications and visualize that the overlap of different categories (cumulative area fractions above 1) is eliminated in the latter.

were dominant at least once. It also shows that abrupt day-to-day changes of dominant patterns are rather rare and a smooth transition from *Gravel* to *Fish* to *Sugar* to *Flowers* is observed.

Overall, the different classification methods agree well with each other and no large discrepancies are found. This reassures that these methods are valid for further analysis of meso-scale patterns. While the $I_{org}$/S metric is computationally cheap and can be easily applied to different regions, it is less suitable for short time periods because the terciles that attribute the $I_{org}$/S pairs to specific quadrants are not robust for a small sample size. Terciles from other studies can only be used if the resolution of the dataset is the same. For short time-periods, like those of typical campaigns where people are interested on specific days

or even sub-daily variations, manual classifications are advantageous as they do not require a large sample size. The neural network approach is a good possibility to extend the classifications to different time periods and regions by using manual classifications as a training set. Here, the manual classifications served as an additional independent validation dataset and proved once more the capabilities of the neural network which has been trained on the dataset of Rasp et al. (2020).

## 5 Conclusions

Meso-scale patterns of shallow convection are a main focus of the EUREC$^4$A field campaign that took place in January and February 2020. To gain a better process understanding of shallow convection in the trades, the classifications of meso-scale patterns offer the opportunity to study the measurements in their meso-scale context and thereby split the observations into less complex pieces and disentangle otherwise superimposed processes. Here, we present C$^3$ONTEXT, a dataset on the common consensus on the meso-scale patterns which occurred during the broader EUREC$^4$A time period and emerges from the manual classifications done by members of the EUREC$^4$A science team.

C$^3$ONTEXT reveals that all meso-scale cloud patterns are observed during the studied period of January and February 2020. However, in the intense observation period of the EUREC$^4$A field campaign, January 20th to February 20th, *Gravel* is only sporadically identified and not prevalent in the study area. In contrast, a week before the intense observation period, *Gravel* dominates in the research area. Instruments that were already running at this time, like those at the Barbados Cloud Observatory, were able to gain measurements under the meso-scale influence of *Gravel* and complement the measurements from the IOP.

The difficulties of the participants to classify the patterns in the output of a storm-resolving simulation demands further investigations into how well simulations can capture the variability of meso-scale patterns of shallow convection in the trades.

A comparison of the manual classification approach with other methods used in the literature to identify the four meso-scale patterns of shallow convection reveals a generally good agreement and confirms the validity of the different approaches. Nevertheless, the manual classifications are beneficial for limited temporal and spatial studies especially when the classifications are done by a group of several trained scientists. It presents a way to gain a consensus of subjectively defined cloud patterns without an additional layer of complexity from a neural network or any other algorithm.

In general, it has been shown that with little effort, classifications of the cloud field are possible and can be a huge benefit for the community, encouraging this approach for future studies.

## 6 Code and data availability

The C$^3$ONTEXT dataset including raw data is openly available at the zenodo database (European Organization For Nuclear Research and OpenAIRE, 2013): https://doi.org/10.5281/zenodo.5979718 (Schulz, 2022a). The source code necessary to generate the dataset is available at https://doi.org/10.5281/zenodo.5989155 (Schulz, 2022b) together with examples on how to process the data and retrieve the classifications for any platform as shown in e.g. Fig. 5.

## Appendix A: Daily classification overview

*Competing interests.* The author declares that he has no conflict of interest.

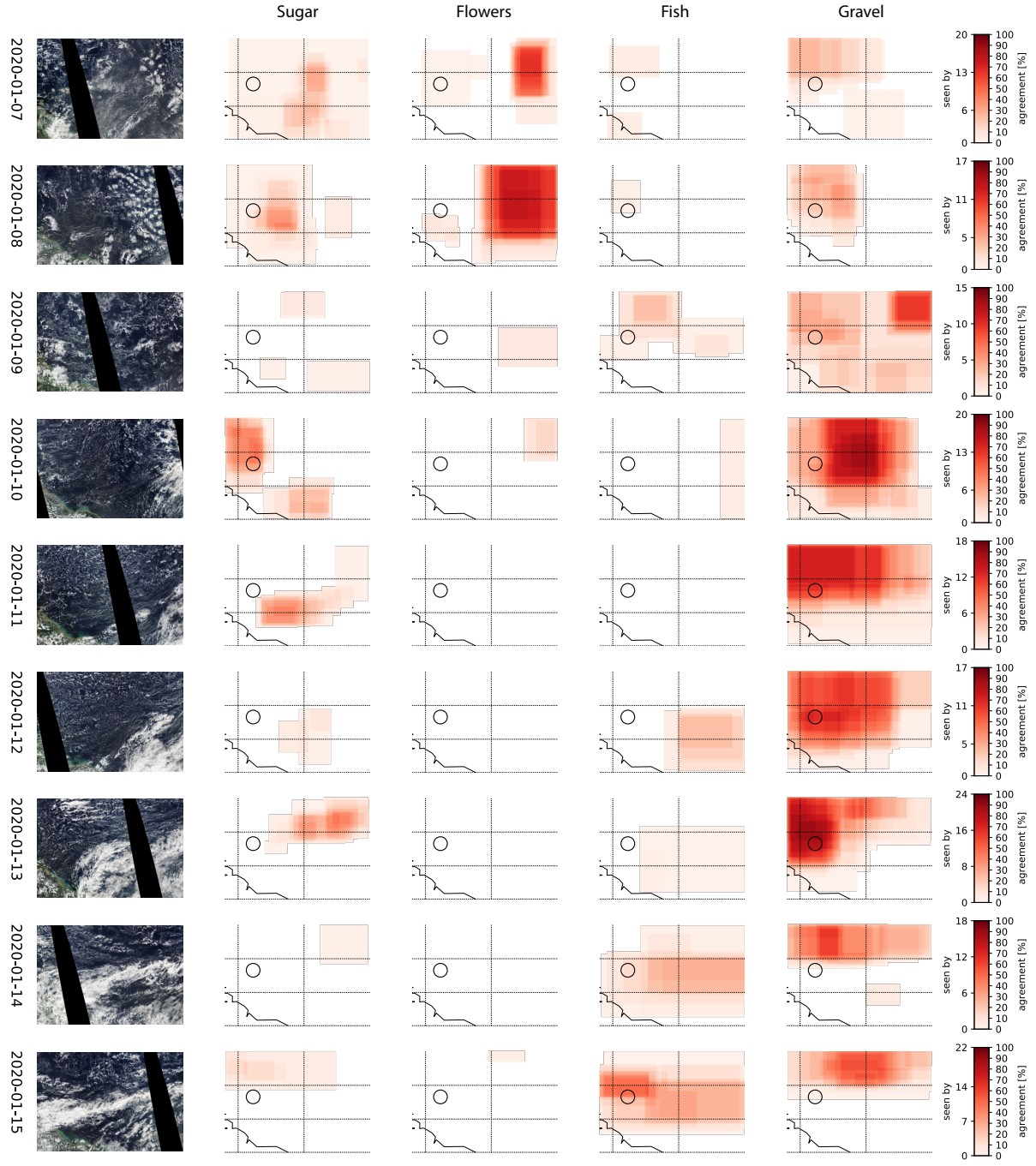

**Figure A1.** Heatmaps of manual classifications based on MODIS (Aqua and Terra) visible imagery. Left to right: Visible imagery during Aqua overpass, User agreement on *Sugar*, *Flowers*, *Fish* and *Gravel* (left to right). The circle marks one focus area of the EUREC$^4$A field campaign and the main flight pattern of the participating research aircraft HALO Konow et al. (2021). Coastlines are based on GSHHG shapefiles (Wessel and Smith, 1996).

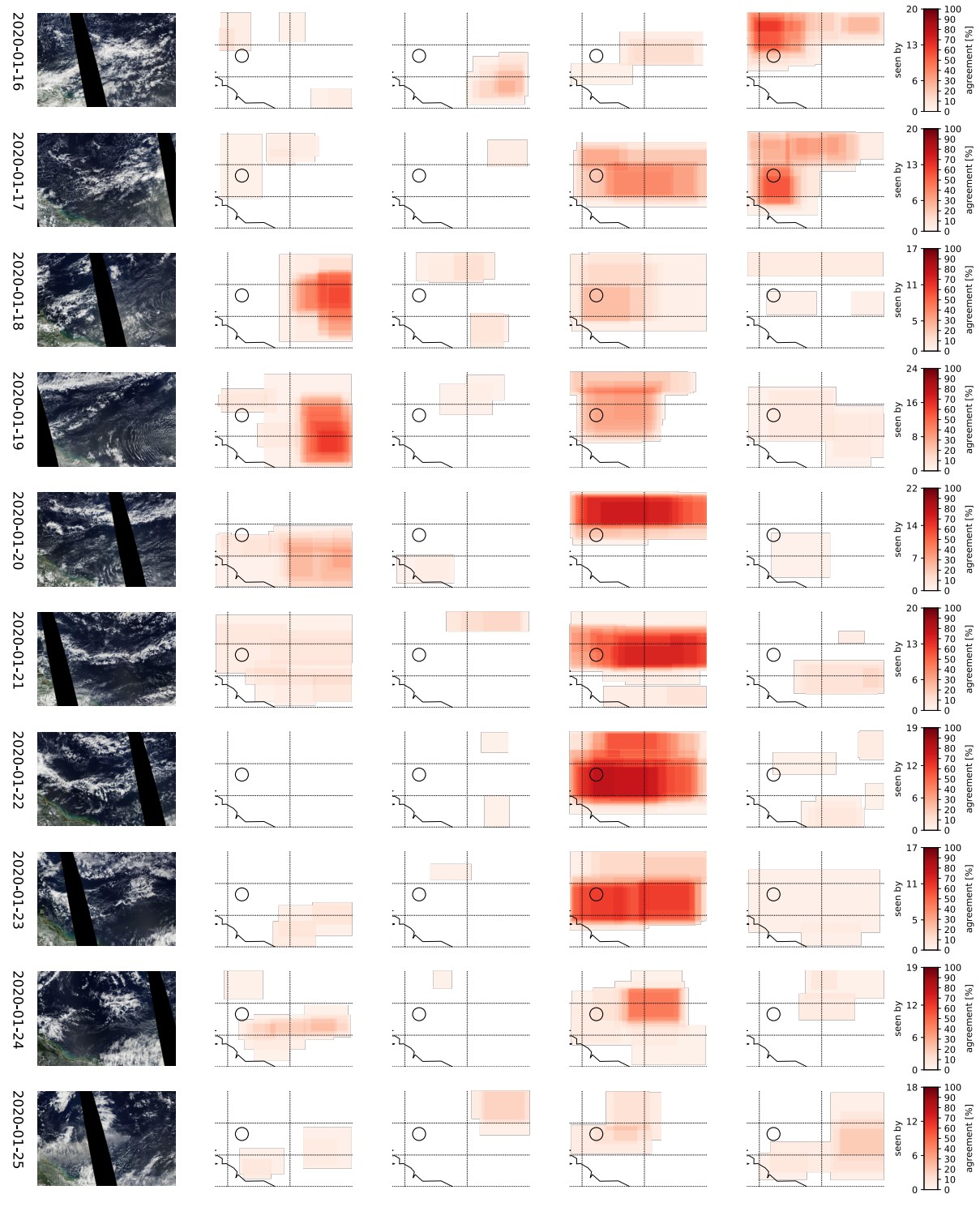

**Figure A2.** Continuation of Fig. A1

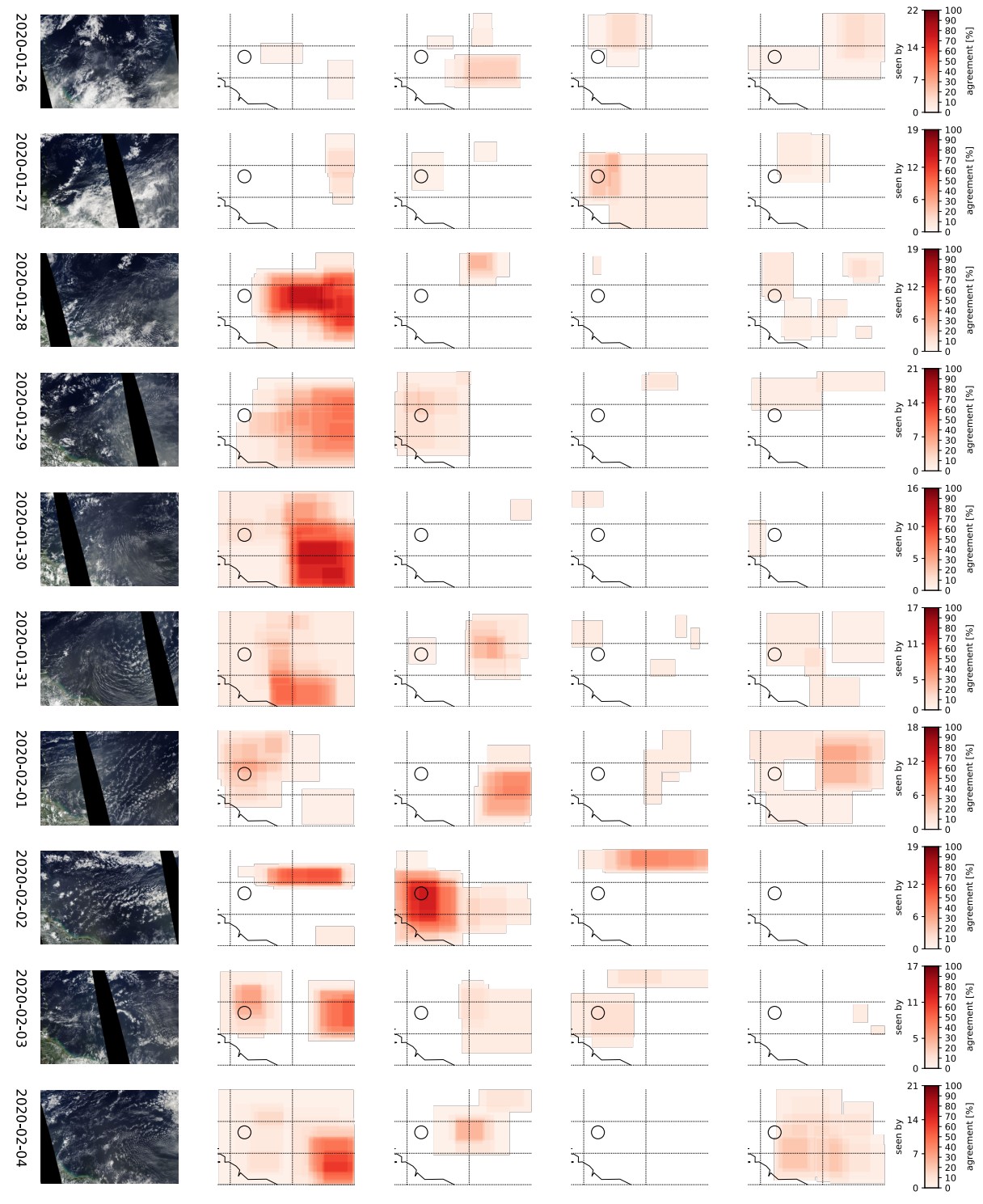

**Figure A3.** Continuation of Fig. A2

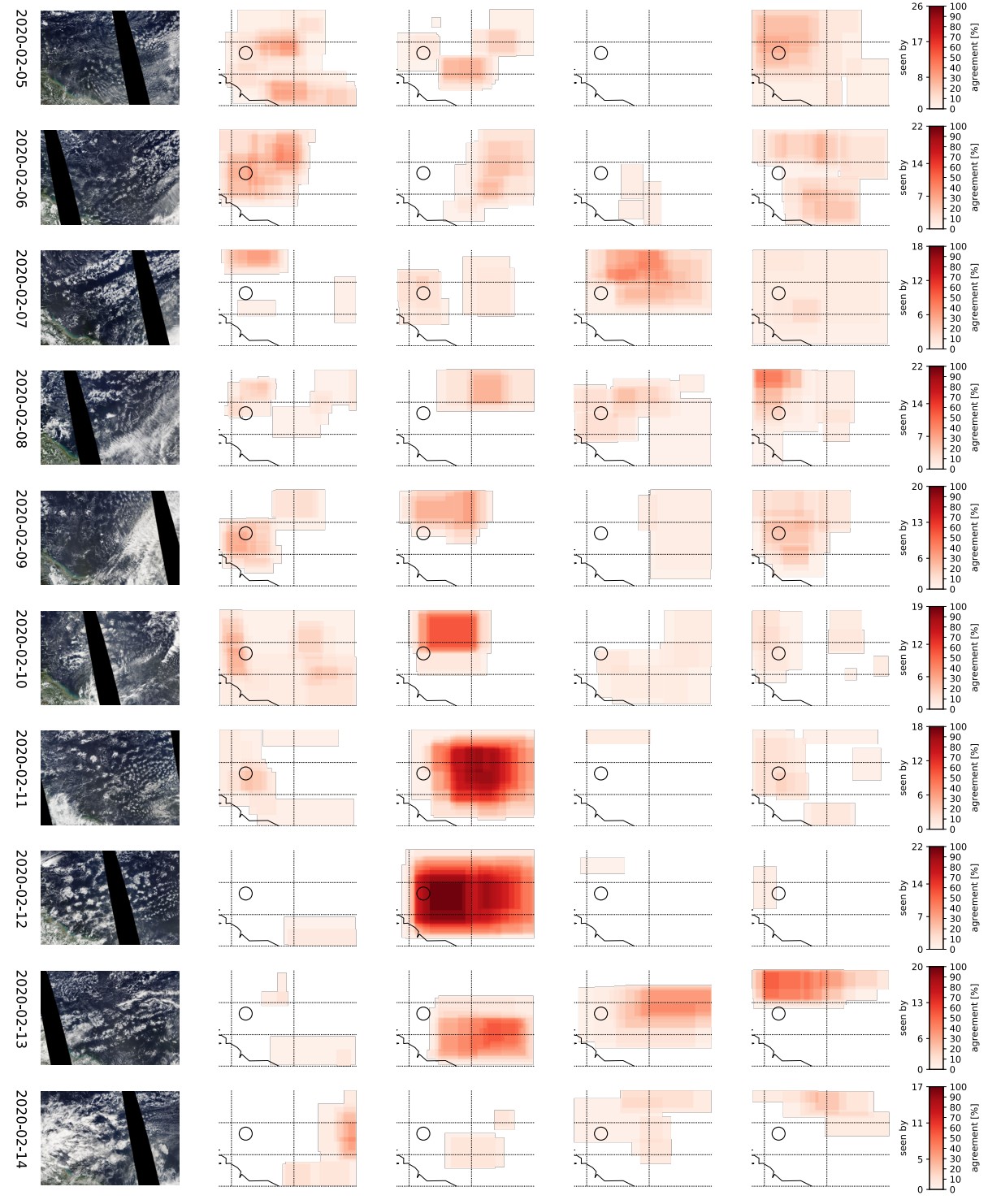

**Figure A4.** Continuation of Fig. A3

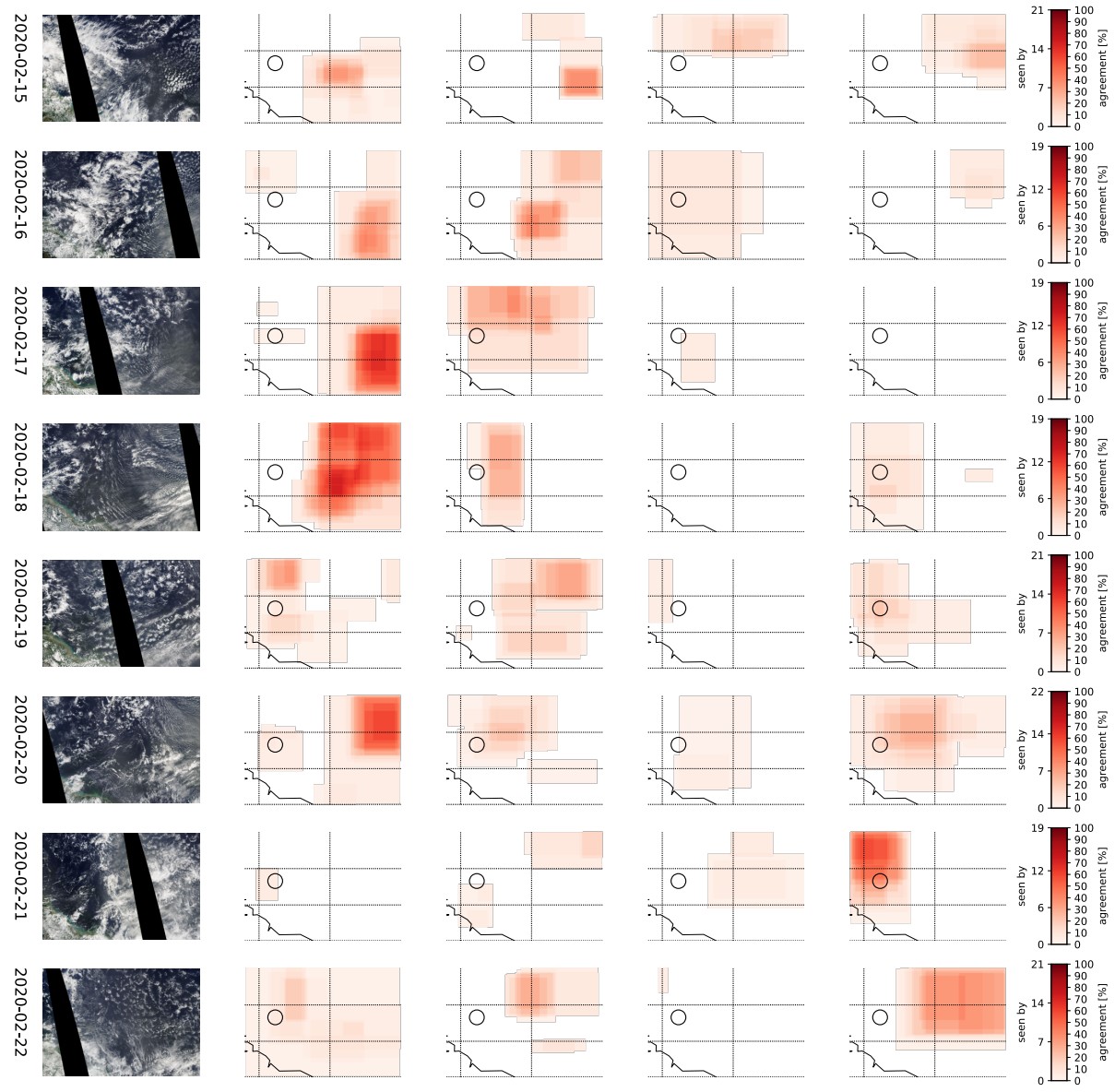

**Figure A5.** Continuation of Fig. A4

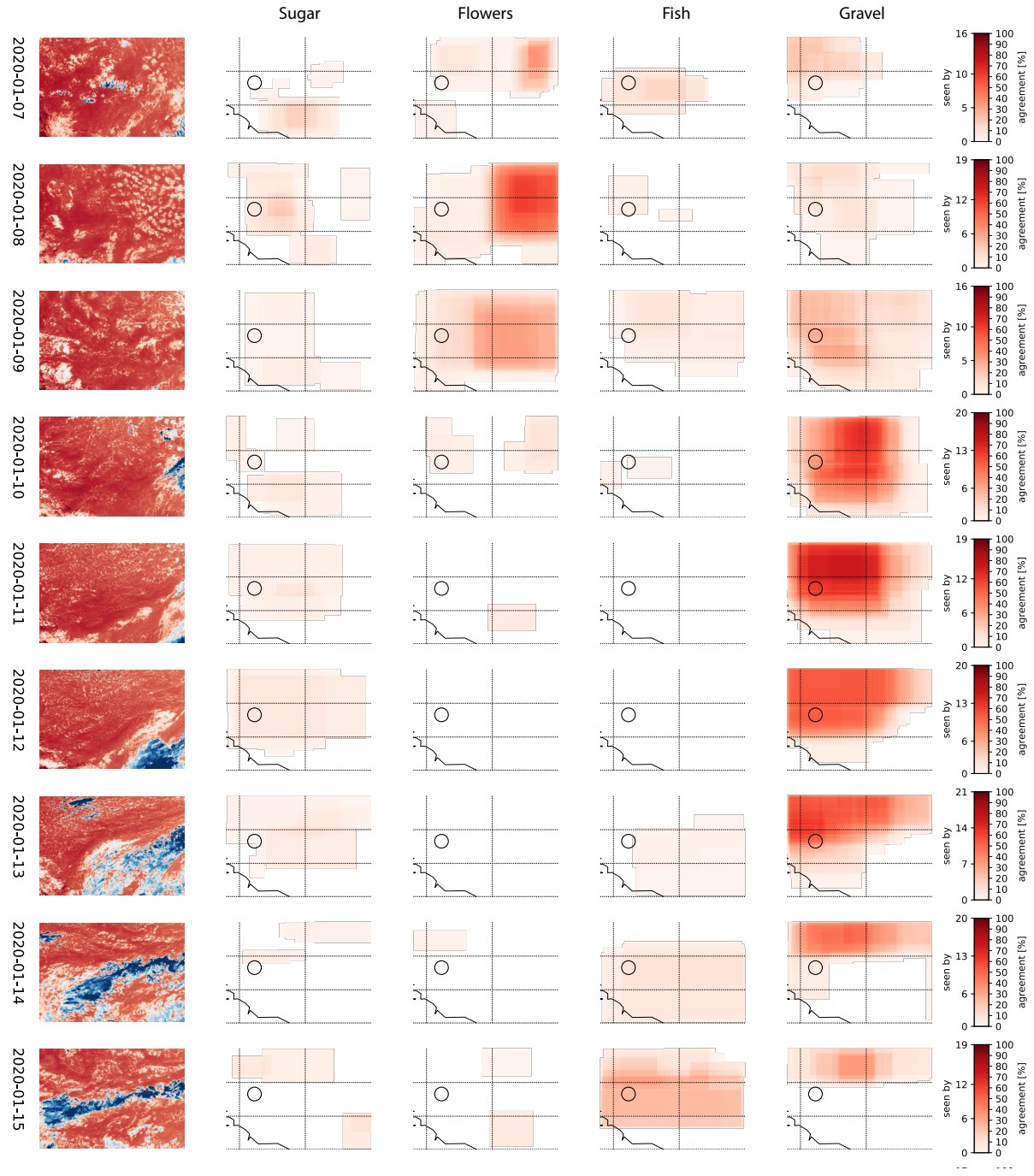

**Figure A6.** As Fig. A1 but for the infrared workflow. Images are showing the cloud field at 16 o'clock. (except 11.02.2020: 17 o'clock)

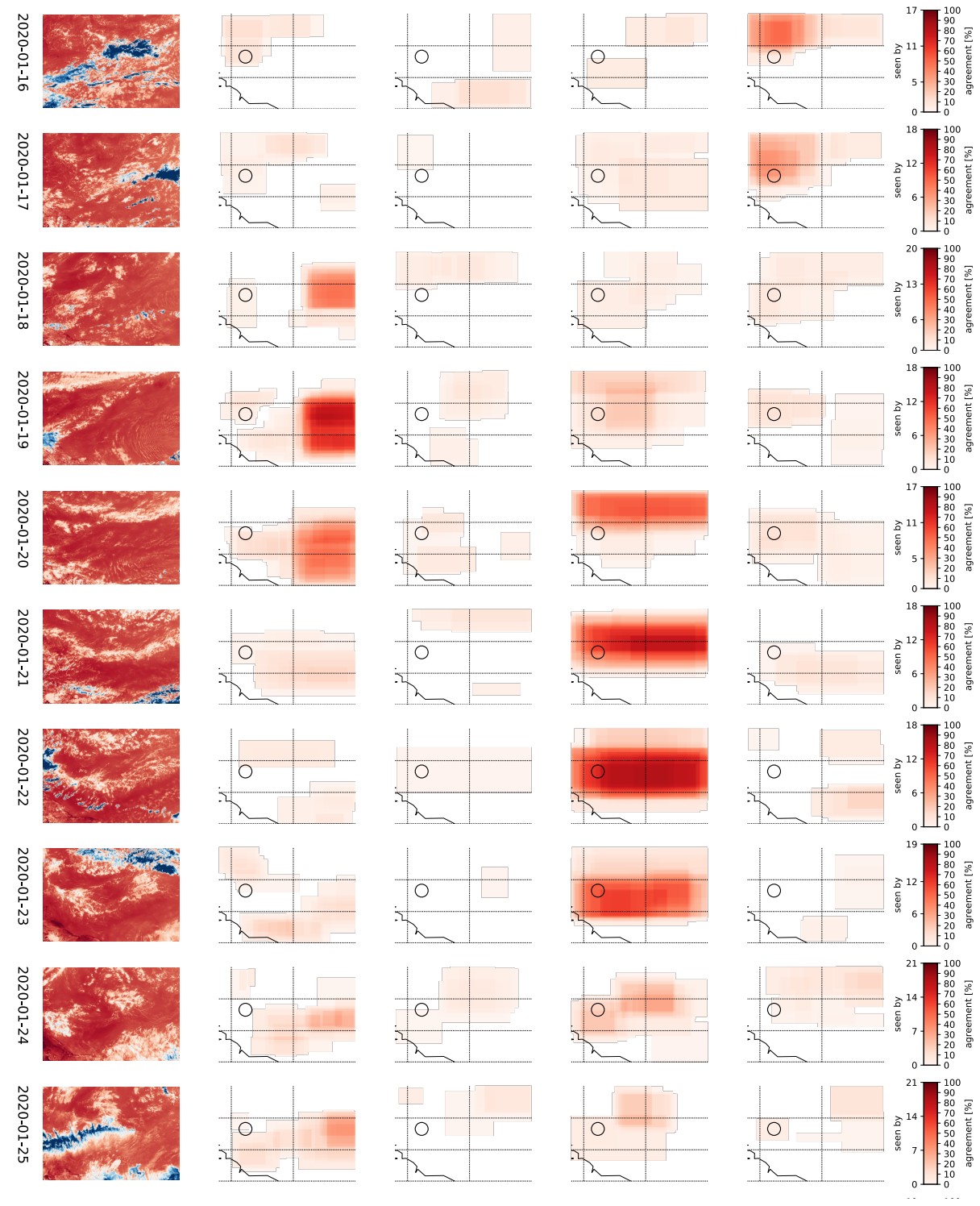

**Figure A7.** Continuation of Fig. A6

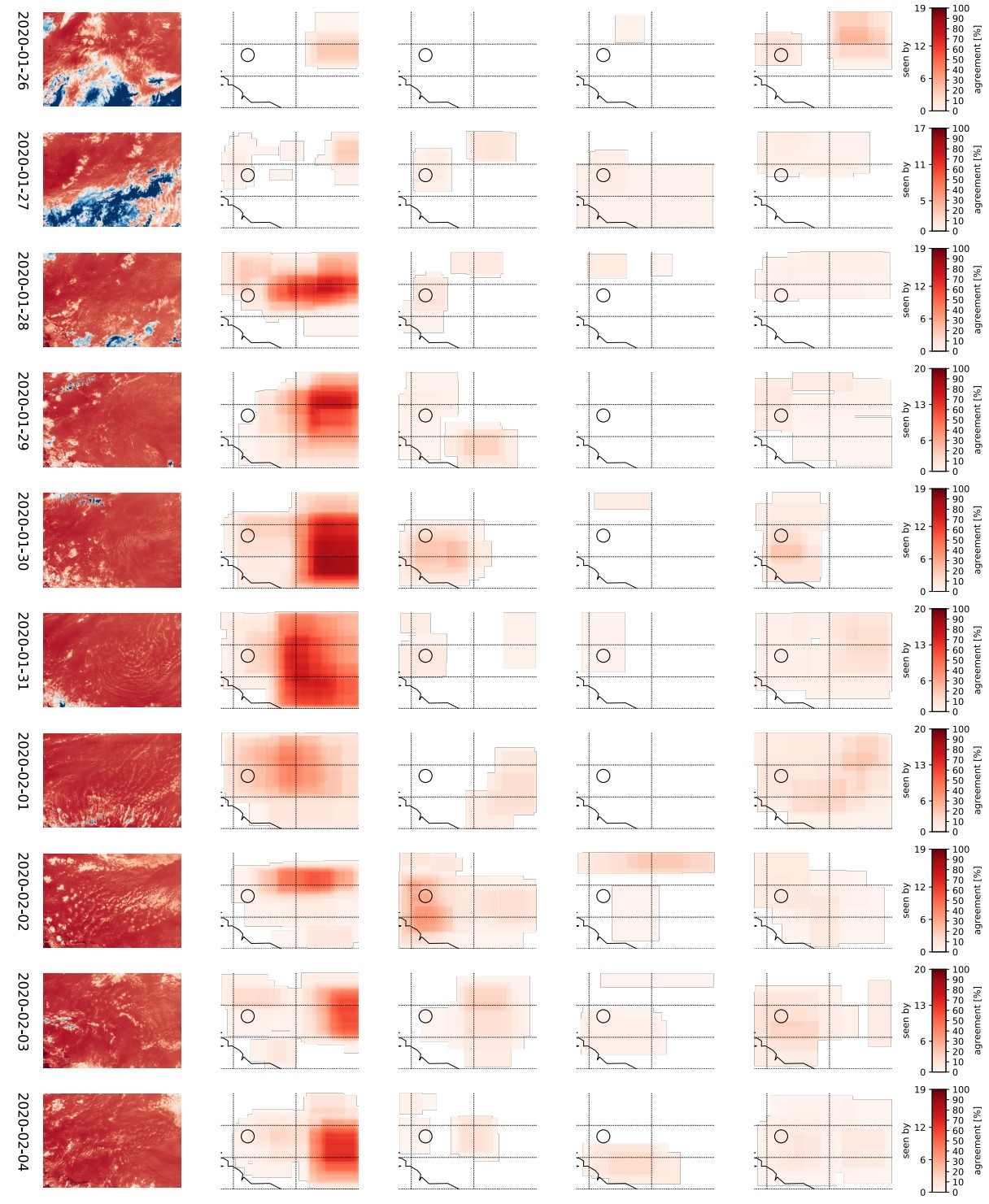

**Figure A8.** Continuation of Fig. A7

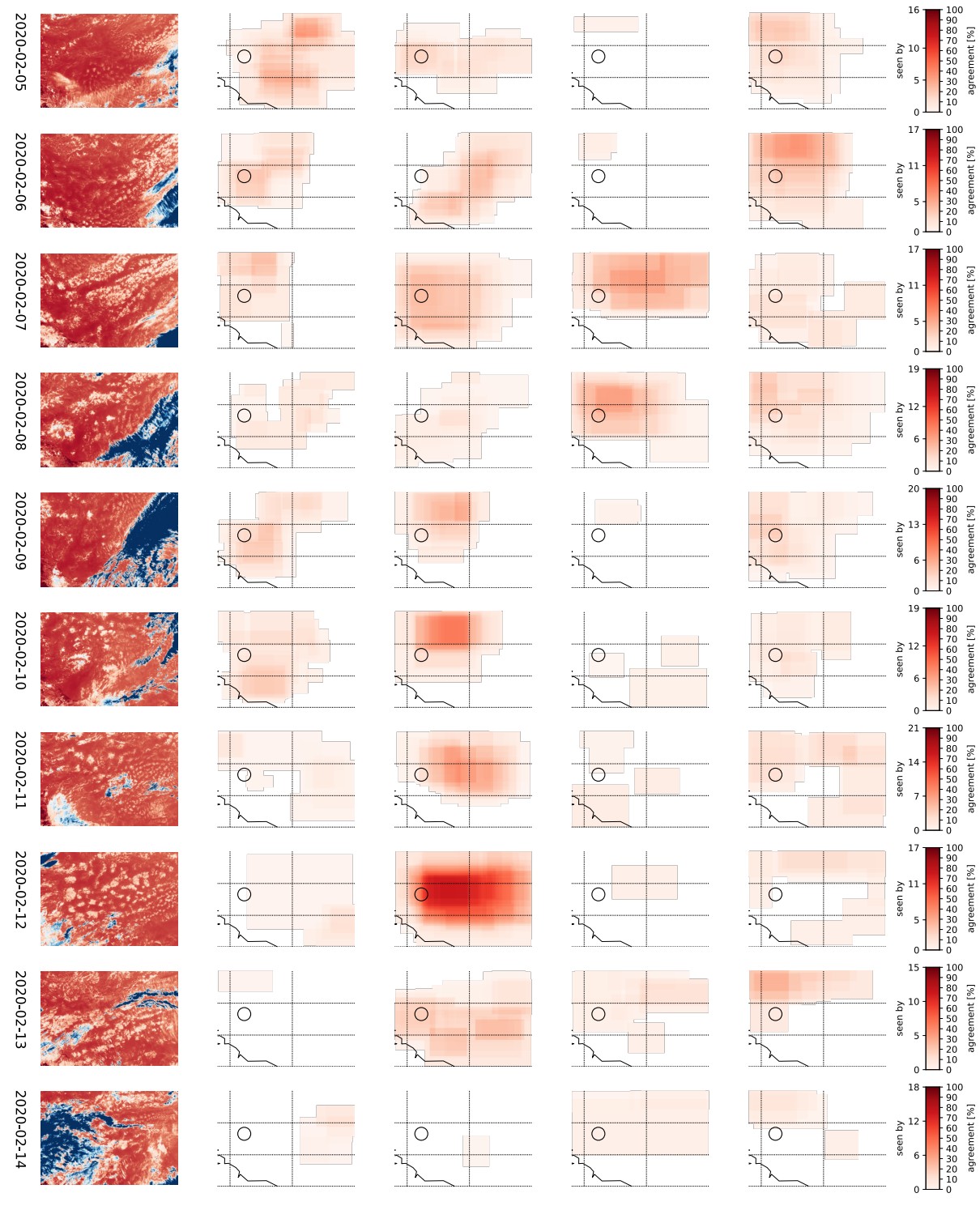

**Figure A9.** Continuation of Fig. A8

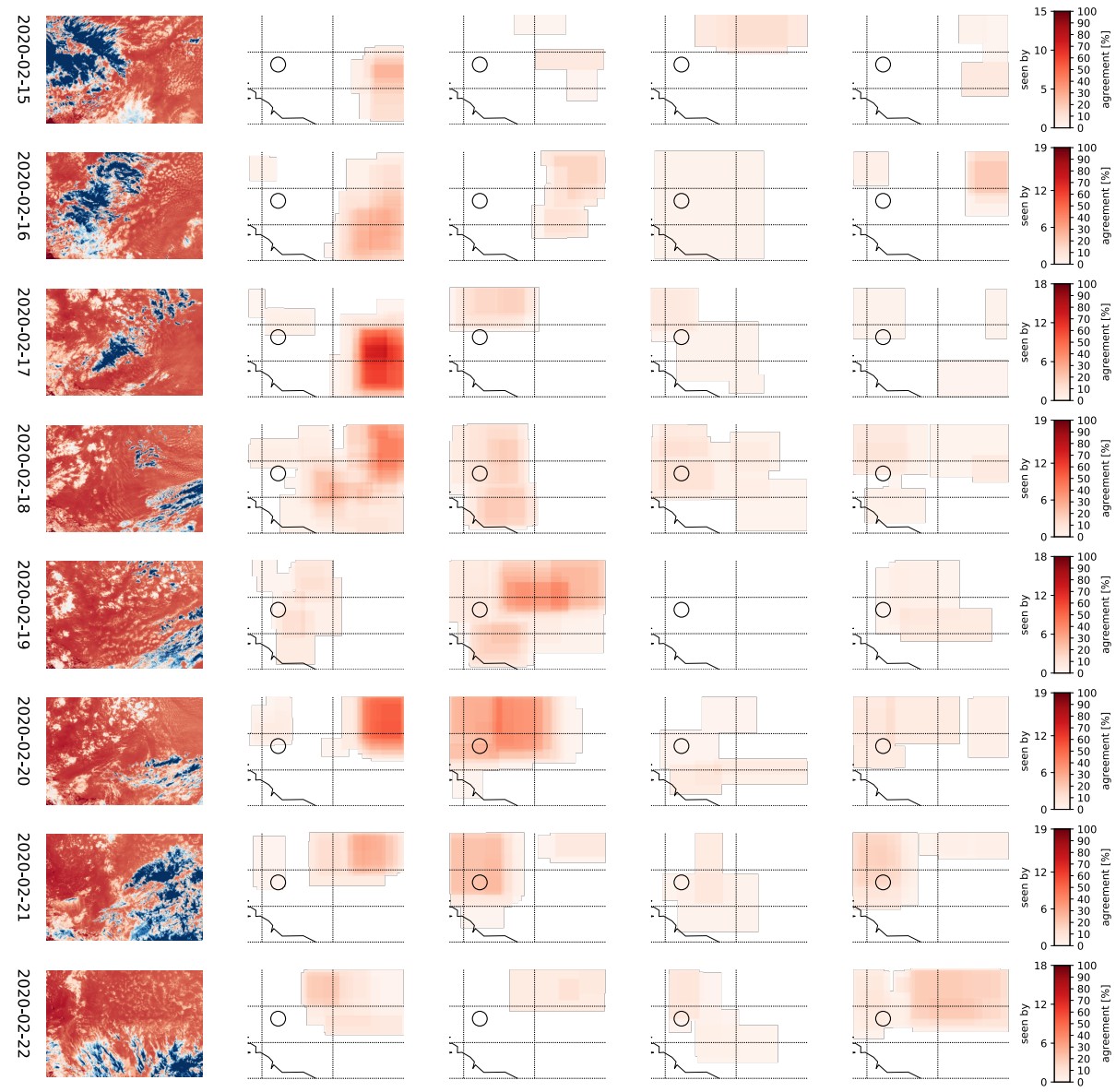

**Figure A10.** Continuation of Fig. A9

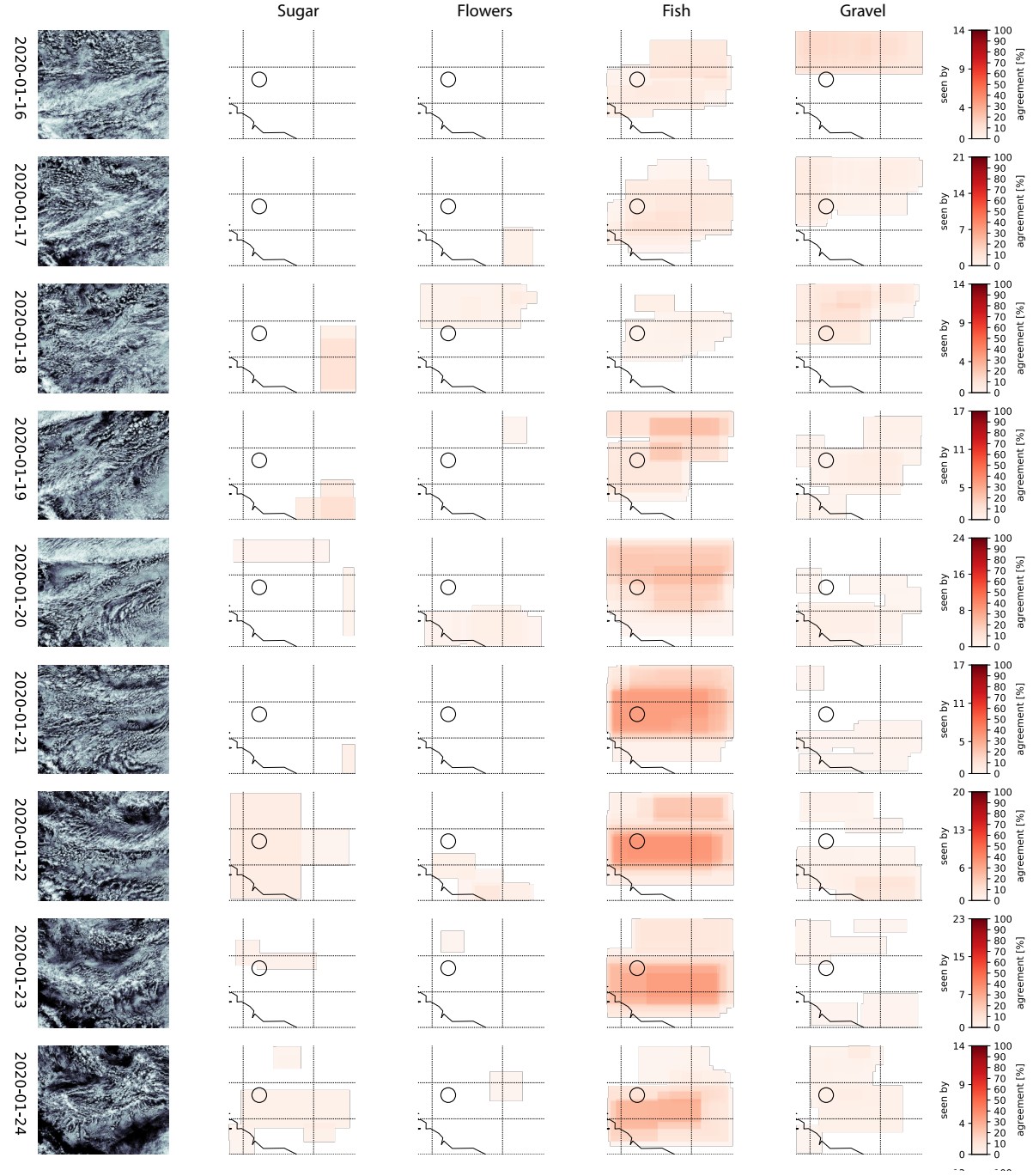

**Figure A11.** As Fig. A1 but for ICON albedo workflow. Images are showing the cloud field at midnight.

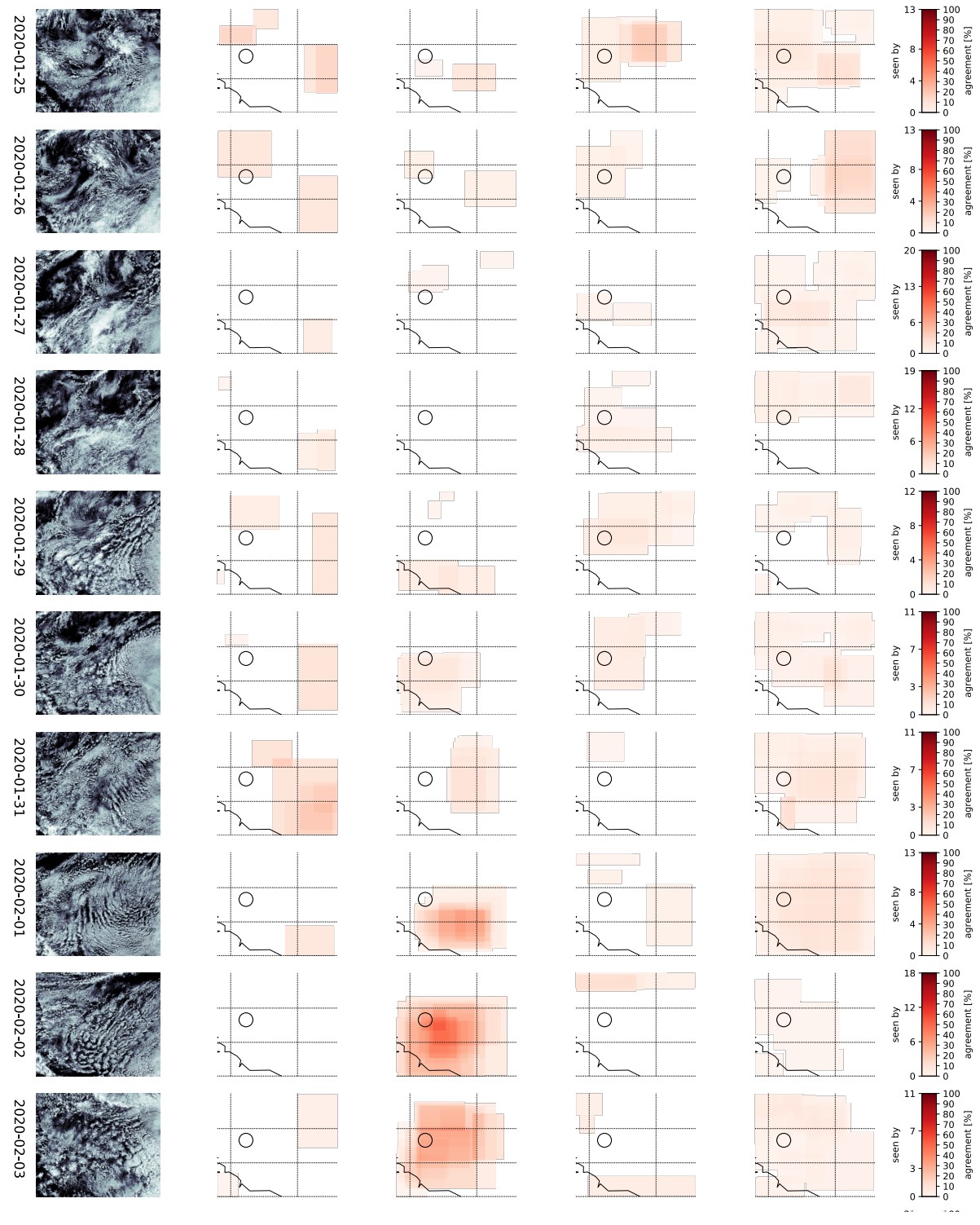

**Figure A12.** Continuation of Fig. A11

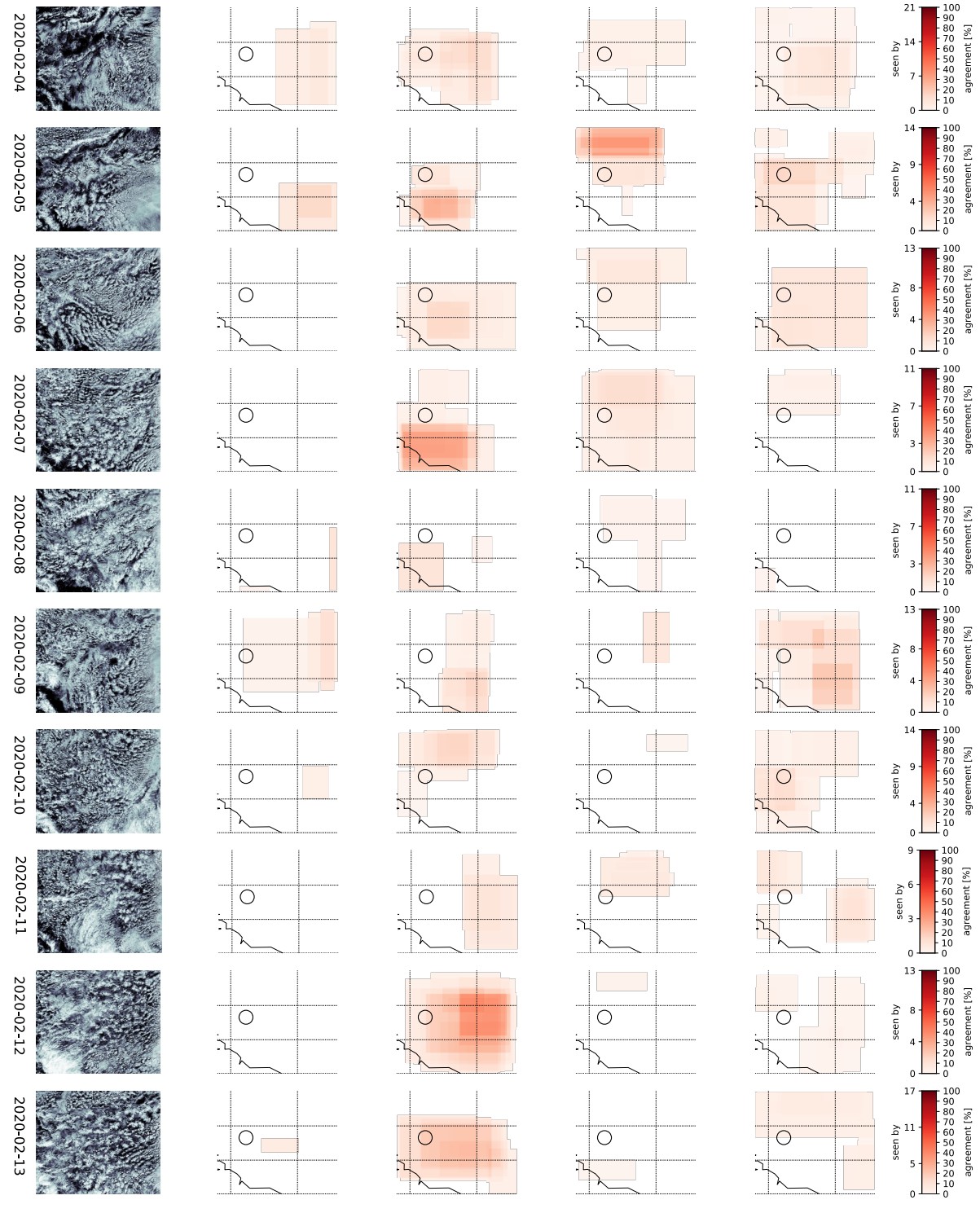

**Figure A13.** Continuation of Fig. A12

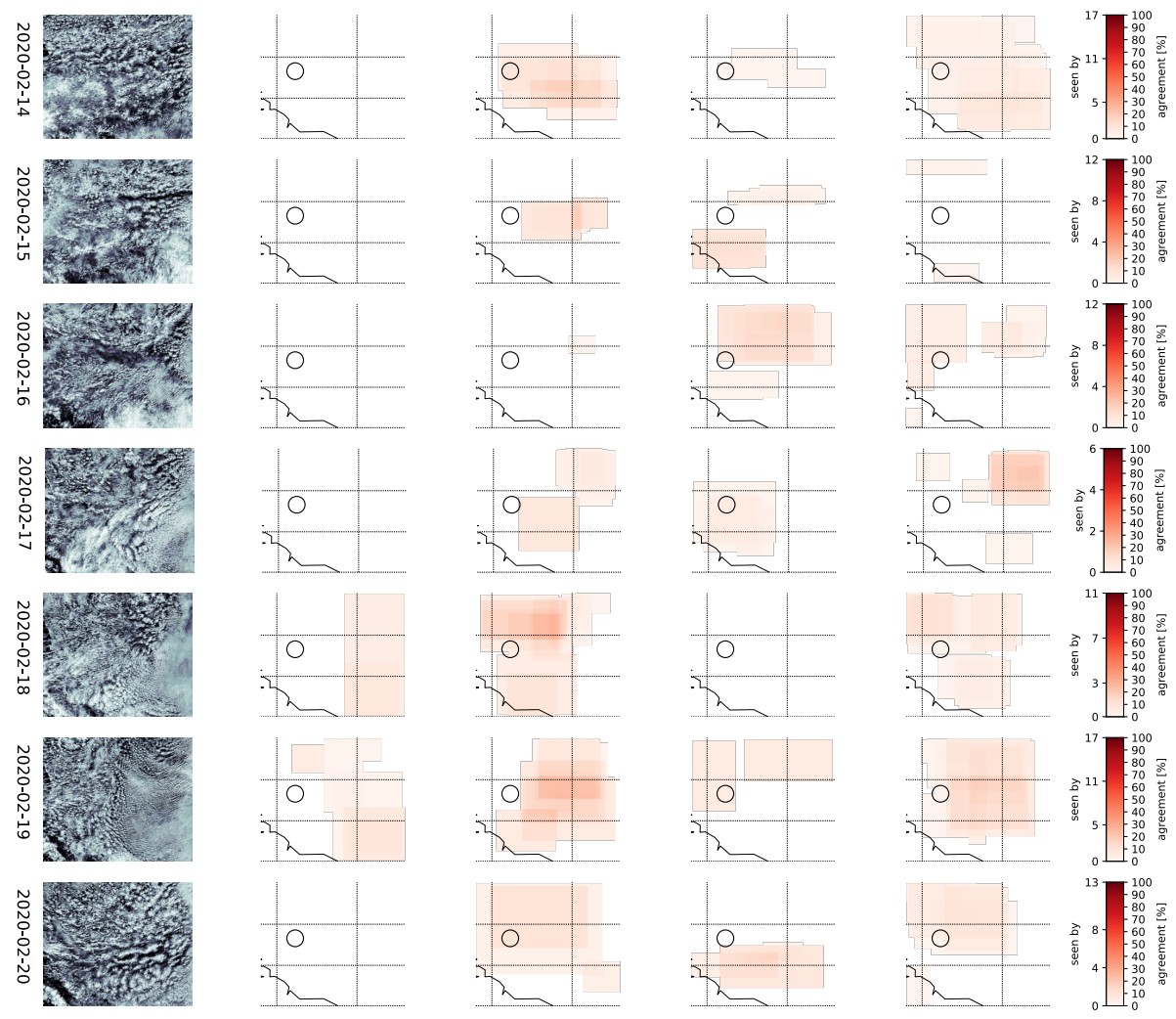

**Figure A14.** Continuation of Fig. A13

*Acknowledgements.* The author thanks the participants of the international remote classification event for their time in labeling the cloud patterns. Daniel Klocke is thanked for conducting the ICON-SRM simulations. Geet George is thanked for helpful feedback on an earlier version of this manuscript. Isabel McCoy, Leif Denby and one additional anonymous reviewer a thanked for their valuable feedback that helped to improve the manuscript. We acknowledge the use of imagery from the NASA Worldview application (https://worldview.earthdata. nasa.gov), part of the NASA Earth Observing System Data and Information System (EOSDIS). GOES-16 Advanced Baseline Imager data is available at https://doi.org/10.7289/V5BV7DSR. Its Level 1b radiances were converted with Raspaud et al. (2019) to brightness temperatures. This publication uses data generated via the Zooniverse.org platform, development of which is funded by generous support, including a Global Impact Award from Google, and by a grant from the Alfred P. Sloan Foundation.

220

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
