# Peer review of "C3ONTEXT: A Common Consensus on Convective OrgaNizaTion during the EUREC4A eXperimenT"

_Earth System Science Data, 2021_

## Author Response (AR1)

**Response to reviewers' comments**

Manuscript: **C$^3$ONTEXT: A Common Consensus on Convective OrgaNizaTion during the EUREC$^4$A eXperimenT**

**Overarching remarks**

I like to thank all three reviewers for taking the time to critically read the manuscript and providing feedback to improve this manuscript. All of the reviewers comments are repeated below and are addressed individually below. Appropriate changes were made to the manuscript and dataset where needed.

**Reviewer 1**

**General Comments**

Schulz succinctly and thoroughly presents a novel, manual identification method and dataset for classifying the four mesoscale morphology cloud patterns thought to dominate the trade wind region. The method utilizes 50 scientists and their knowledge of cloud morphologies to build up a robust dataset of 2-hourly identifications based on satellite data over a two month period that overlapped the recent EUREC4A field campaign. These identifications are designed to give context (hence the clever acronym) to the measurements performed by the plethora of platforms available during the campaign. In addition to presenting the methodology of the identifications, Schulz details the various levels of data that are available to work with (including providing example scripts on GitHub) and demonstrates several ways that these identifications can be used. A particular asset in this paper is the section comparing the manual classifications to previous methods of identifying these cloud types, clearly demonstrating the consistency in identification of the manual method with the more computationally difficult neural network and Iorg/S methods. The effort and thought involved in developing this dataset is commendable. Overall, the manual identifications will be a useful unifier for EUREC4A measurements and will facilitate more precise comparisons between campaign-based and other studies on these cloud types. The detailed presentation of this dataset here will be an excellent resource for the community.

**Specific Comments**

- ✅ Intro/Figures (see technical comments): What is the actual domain that you are covering in the manual identifications? It would be helpful to explicitly call out this region (or regions if you change where you are focusing, e.g., Figure 7) and how it relates to the general region covered by EUREC4A.
  **Response**
  The domain covered by this dataset has been detailed in Tab. 1. The information is now also added to the text and contextualized in the overall EUREC4A campaign domain foci (Tradewind alley and Boulevard des Tourbillions). In addition parallels and meridians are added to Fig.4 and the overview figures in the appendix.
- ✅ Intro: I appreciated the inclusion of the ICON simulations in this effort and find your teaser for later work on this comparison compelling (e.g. sections 2 and 5). However, I did find its integration with the rest of the paper a little abrupt. It might help to explicitly mention in the introduction that you will be looking at high resolution simulations through ICON (it is a surprise when it first appears on line 46) and briefly explain why.
  **Response**
  The usage of storm-resolving simulations is now also mentioned and justified in the abstract and introduction. The analysis has been slightly extended as well following the comments of Reviewers #2 and #3

- ✅ Figures 5 and 6: In general, I think you use level 3 data to produce your analysis (you have it in the GitHub script for Figure 5). It would be helpful to explicitly state in the text what form of the data you are using in your analysis (as you do for Figure 7) for reproducability.

  **Response**

  As the reviewer correctly mentions, the reproducability of the shown Figures is already given by the additional material (see Code and Data availability section). We mention however the process-level of the data used for Fig. 5 now explicitly, because the reader might want to create a similar Figure for a different platform or location.

- ✅ Line 162: You say the manual classifications are "naturally more accurate"... how do you reach this conclusion and can you expand on this logic in the text?

  **Response**

  The paragraph has been rewritten to also incorporate the comments of Reviewer #2 and to clarify this causality.

**Technical Comments**

- ✅ Line 25-32: Worth including the region that you will be examining in the manual classifications and how it relates to the EUREC4A study region.

  **Response**

  The information has been added to the manuscript

- ✅ Line 56: Did you have 50 or 51 scientists? Varies across manuscript.

  **Response**

  50 scientists participated. This has been made consistent throughout the manuscript now.

- ✅ Figure 4: Suggest noting the lat-lon dimensions in the caption. It would also help to label the sugar, gravel, flower, fish columns explicitly in this first figure at least (not necessarily for all the appendix ones) for easier use.

  **Response**

  Figure 4 and the figures in the appendix have been updated. The individual columns are now labeled and meridians and parallels are added for a better orientation. The colorbar has been adjusted to only show colors in case any classification has been made. In the last version a slight red marked values of zero.

- ✅ Line 117: there are more than three platforms involved in the campaign. Suggest removing "the" in front of "three" and referencing Stevens et al. 2020 for a list of all the platforms involved.

- ✅ Figure 5: Suggest also labeling plots with the platform names.

  **Response**

  Figure 5 has been adapted accordingly.

- ✅ Figure 7/Line 154-155: Is this the whole domain of the manual classifications? If not, why have you chosen this subset? It looks like it is a little different than the ones used in Bony et al. 2020 and Schulz et al. 2021.

  **Reponse**

  There has been a typo in the noted extent of the domain. This typo has been corrected. The domain is the same as in Bony et al. (2020) and Schulz et al. (2021).

**Reviewer 2**

**General Comments**

Schulz clearly and concisely presents C3ONTEXT, a dataset quantifying the meso-scale cloud patterns manually identified during the recent EUREC4A field campaign. This data set is innovative and calculated with great care, and its results are compellingly presented. The data set will be of use to provide a standardized view of the cloud organization state during EUREC4A to augment measurements from other coincident platforms. Schulz also shows that results from this manual classification compare well with results from two commonly used

approaches, which is another useful aspect of this manuscript.

**Specific Comments**

- ✅ Could you also clarify how many times a scene was classified as 'other'? This information is interesting as well for quantifying how comprehensive these four patterns are, how often patterns are hybrid / or in transition from one pattern to another, etc. This information is implicitly present, e.g. in Fig. 5 and Fig. 7, but could merit a brief but explicit clarification. Someone not familiar with these patterns might jump to the conclusion that these patterns are nearly always present.
  **Response**
  The scenes have not actively been classified as 'others' but were rather left unclassified. Nevertheless, this information might be indeed useful. Fig. 5 and Fig. 7 have therefore been adapted to also reflect the amount of the unclassified area (Fig. 7) and times users did not attribute any pattern to a specific location (Fig. 5).

- ✅ The comments about storm-resolving models felt a bit interspersed throughout the manuscript, and it might merit a very short subsection bringing these results together. While not the focus of this manuscript, these comments are interesting, such as line 65."This supports the assumption that larger features are better reproduced in storm-resolving simulations than features of smaller scales, like Sugar", and foreshadow future work using these models.
  **Response**
  The classifications based on the storm-resolving simulations are now also mentioned in the abstract and introduction to follow the advise of Reviewer #1 to integrate this part more into the flow of the manuscript. The comparison with the other workflows is now more detailed by including the workflow e.g. also in Fig. 7.

- Figure 6:

  - ✅ It would be useful to add information to Fig. 6 to ease comparison with the Bony et al 2020 Iorg/S quadrants. In Line 150, you label the quadrants (e.g. Fish in the upper right corner), and perhaps such a labeled grid could be added to Fig. 6.
    **Response**
    Fig. 6 has been updated for a better comparison with Bony et al. (2020). The unit of the mean cluster size is now identical. The methodology used in Bony et al. (2020) to calculate the quadrants has been applied as well. However, it should be noted that this calculation depends on the sample size and the spatial resolution of the satellite images. The results are therefore best compared to Fig. S4c of Bony et al. (2020), where the same satellite product has been used.

  - ✅ I also find the symbols a bit small and sometimes hard to distinguish the colors/patterns. The figure is, however, striking regarding the good agreement between the manual classification and the deep neural network.
    **Response**
    The size of the symbols and the figure itself have been increased for better readability. The colors are left as is, because they are consistent with e.g. Schulz et al. (2021) and Stevens et al. (2020) and are only dublicating the pattern information that is already encoded in the orientation of each wedge.

  - ✅ Could you elaborate briefly on the lack of patterns identified in the bottom right corner? e.g. why flowers are more centered (line 150)?
    **Response**
    Flowers, especially those that we detected during the campaign do not separate that well from the other patterns. This might be improved by using different metrics as suggested by Janssens et al. (2021). Flowers need to be quite large to separate well. The "missing patterns" in the bottom right can also be seen in Bony et al. (2020; their figure S4). The manuscript has been adapted to include this clarification.

- ✅ Figure 7: Can you explain why the deep neural net seems to identify a higher area fraction covered by each pattern? It also seems like there is a higher area fraction covered by each pattern for the IR (center bar) than the visible (left bar)

  **Response**

  The higher area fractions of the deep neural network classifications and the infrared are caused by their temporal coverage. Both the neural network and the manual infrared-based classifications cover the complete diurnal cycle and are therefore able to capture the mesoscale context even at night, when some patterns preferentially occur. (see Vial et al. (2021)). Fig. 7 has now been adapted to show only the time period between 12 and 20 UTC where all classification workflows overlap. A clear separation between the different workflows is no longer recognizable.

- ✅ Could you clarify the smallest rectangle an identifier can use? Or are the rectangles of fixed size?

  **Response**

  There has been no limit on the size of the bounding box a person could draw. The only upper limit has been the size of the given image itself. For clarification, the manuscript has been adapted.

  > …drawing rectangles of variable sizes…

- ✅ It might be worth considering contextualizing the size of the domain for manual classifications with the two main stages of EUREC4A: Boulevard des Tourbillons and Trade-wind alley, and possibly the EUREC4A circle and ATR rectangle.

  **Response**

  A similar note has been made by Reviewer #1. The main regions of interest of EUREC4A are now mentioned along with with the coordinates of the domain directly in the introduction.

**Technical Comments**

- ✅ Line 120. Small typo Boulevard des Tourbillons (and extra m). Might give translation (eddy boulevard) as well.

- ✅ Clarify 50 vs. 51 (on line 55) researchers

  **Response**

  50 scientists participated. This has been made consistent throughout the manuscript now.

The acronym is fantastic!

**Response**

Thank you very much!

**Reviewer 3**
* * *
Schulz presents an in-depth and well-written description of a very useful dataset of manual classifications of shallow cumulus clouds covering the EUREC4A campaign period. The comparision with more traditional techniques for measuring organisation provides a valuable reference for interpreting the manual classifications. And contrasting the results of using different datasources is very valuable. The detail with which the dataset is described and the openeness by which the tools used have been shared is commendable and inspiration for our community as a whole.

There are two aspects I would find very valuable to giving a little more depth. First, is the definition of "truth" in this manual classifications. This would focus on answering questions such as "which of workflow dataset should we trust as the most truthful and why?" "is it possible to produce a kind of consensus among the four workflows?" This could draw in prior studies using manual classifications referenced in this publication. Second, currently there is little analysis of the manual classifications created for the simulation data (as compared to the observation-based workflows), why is this and was this analysis done?

**Response**
Thank you for taking the time to review my manuscript and providing me these useful comments. Before responding to the detailed comments, I want to address the two above raised aspects.

1. The request to provide further guidance on the classification workflows and which one should be used or regarded as truth is reasonable. In general, it can be said that the workflow based on the infrared images is the most versatile one and can be used to contextualize measurements on both the sub-daily and daily time-scale. Qualtitative differences between the visible and infrared workflows could not be identified. For convenience, the infrared workflow might therefore be chosen in most cases. In order to prevent introducing biases in e.g. the daytime versus nighttime classifications, the different workflows are not combined, but the usage of the infrared dataset itself is encourage now explicitly in the manuscript.

2. The analysis of the classifications of the simulation output has clearly been neither the focus of this paper nor of the classification event itself. The classifications of the simulations rather serve as a way to emphazise the importance to further understand the processes leading to the different cloud patterns as they are not all represented in the conducted storm-resolving simulation. This purpose has been emphazised and finds itself now also in the abstract and introduction. Nevertheless, Fig. 7 has also been extended to include the time-series of the manual classifications of the ICON run to compare it with the observational ones more closely.

**Detailed comments**

- ✅ [p 1 | 21] "studies concentrated on the classification of meso-scale patterns…":

  - You could also mention Denby 2020 here as an example of using a neural network for classification. It would also be good to mention Wood & Hartmann 2006 here already. You should also include reference to Janssens et al 2021 (https://agupubs.onlinelibrary.wiley.com/doi/full/10.1029/2020GL091001 (https://agupubs.onlinelibrary.wiley.com/doi/full/10.1029/2020GL091001)) for it's review of traditional metrics

  - **Response**
    Reference to Denby (2020) has been added. Janssens et al. (2021) is referenced later in the manuscript (see comment on [p 8 | 132])

- ✅ [p 3 | 59] "as it quickly turned out that the identification of the patterns in the model simulation was too demanding. The features had too little similarity with those found in nature.":

  - I think this needs reformulating. "Accumulated intentionally" isn't so clear, you mean that people tended to label the observations and not the simulation output? You could refer to the totals in Figure 2 to make this point. I would also emphasise that you are continuing to talk about classifications made in the ICON workflow in the following sentences.

    Because the you lead the paragraph with the total number of classifications I initially read this as if "sugar" was hard for people to identify across all workflows (but that isn't the case I think, cf Figure 2)

  - **Response**
    The paragraph has been written to incorporate the suggested changes.

- ✅ [p 5 | 95] "This process eliminates overlaps of same-user classifications for each pattern and turns the data into masks, rather than coordinates (see Fig. 3)":

  - What happens if the user draws two bounding-boxes with different types of classification that spatially overlap? (It shouldn't really happen, but maybe you could

just state that it didn't)

   ○ **Response**
Users were able to draw overlapping bounding boxes. In case these boxes were of:

- the same pattern, the overlap is counted only once by simply calculating the union of these bounding boxes

- different patterns, the overlapping region is counted towards both patterns. This case is not handled specifically as it can be used to estimate the uncertainty a user had to classify a specific region as one or the other pattern.

- the following has been added to the manuscript:

> In cases where same-user classifications of different patterns overlap, the overlapping region is counted towards all classified patterns. This case is not handled specifically as it shows the uncertainty a user had to classify a specific region as one or the other pattern.

- ✅ [p 6] "Figure 4":

   ○ There is a bit of aliasing for the text in figure 4. Maybe storing as a pdf/svg would be better?
**Response**
The figure is now embedded as pdf and should now also look smooth when zoomed in.

- ✅ [p 8 l 125] "Comparison with other classifications":

   ○ There doesn't seem to be any analysis of the ICON simulation classifications. It would be good to mention why this was left out.
**Response**
As mentioned in the response to the general comments, the classifications of the ICON simulations have not been the focus of this manuscript and should be seen as supplemental material. Nevertheless, the analysis has still been slightly extended and better integrated into all sections.

- ✅ [p 8 l 132] "org, Tompkins Adrian M. and Semie Addisu G. (2017)) with the mean cluster size (S)":

   ○ In choice of these metrics maybe you could mention how this fits with the findings for Janssens et al 2021?

   ○ **Response**
The findings of Janssens et al. (2021) are now mentioned in the manuscript as:

> Although Janssens et al. (2021) show that different metric combinations, like cloud fraction and fractal dimension, better describe the variance in a cloud field, the pair of Iorg and mean cluster size have been widely used and are considered here for better comparison.

- ✅ [p 8 l 136] "we focus on a domain size of 10×10 degrees to do the comparison":

   ○ I would rephrase to be clearer to something like "we calculate this metrics over a 10x10 degree sub-domain". I would say "Specifically" rather than "Precisely" next. Also 10N to 15N is only 5 degrees, should it be 5N to 15N?

   ○ **Response**
- The domain extends from 10-20 N. This has been corrected in the manuscript:

> Because the Iorg/S measure is sensitive to the domain size, we compute these metrics over a 10x10 degree sub-domain and consider only classifications within this domain for the comparison. Specifically, we focus on the region 10N - 20N and 58W - 48W.

- ✅ [p 8 l 134] "detect the patterns in geostationary infrared images of GOES-16 ABI (Schulz et al., 2021)":

○ Does this mean that the network was trained on the dataset from the EUREC4A IR (channel 13) workflow to predict the manual classifications (masks) created by participants in the same workflow? It would be helpful to emphasise this, and specifically in say the caption of Figure 7, to so that the neural network was trained on the IR data (which would also explain why the agreement is better with the IR rather than the visible manual classification)

○ **Response**
The deep neural network has not been trained with the manual classifications presented here. The time-period of the EUREC4A field campaign is unknown to the network. This has been clarified in the manuscript:

> It should be noted, that this deep neural network has not been trained with the manual classifications presented here, but with the manual classifications of Rasp et al. (2020) which captured older years and included different regions. The network is identical to the one used in Schulz et al. (2021).

I followed the suggestion to include this information also in the caption of Fig. 7.

- ✅ [p 8 l 144] "we expect Gravel and Flowers to be rather regularly distributed and therefore to have a lower Iorg compared to Fish and Sugar":

  ○ I don't quite understand why "flowers" should be more "regularly distributed" than "sugar"? My understanding is that "sugar" is scatter small cumuli which I would expect to be very regularly distributed.

  ○ **Response**
    ▪ By visual inspection, *Sugar* would indeed have a lower degree of organization than *Fish* and *Flowers*. However, as written in the manuscript, the analysis is done based on the measured brightness temperatures, which need to be thresholded to decide between clouds and clear-sky. Because *Sugar* clouds do not have a large vertical extent, only a few clouds are detected as such while the majority is too low to overcome the threshold. The manuscript reads now:
    > It should be noted that Iorg is calculated based on a threshold in brightness temperature and therefore only the deeper clouds in the *Sugar* field are detected leading to a higher Iorg than one would expect from a rather randomly distributed cloud field.

    ▪ This is in agreement with Bony et al. (2020) who introduced this methodology and regarded only the deeper clouds in case of *Sugar*

- ✅ [p 8, Figure 6] "mean cluster size (S)":

  ○ What are the units of "mean cluster size"? Are they pixels?

  ○ **Response**
  The unit of "mean cluster size" is dimensionless. The mean cluster size is given as fraction of the domain. A mean cluster size of 0.3 x 0.002 is therefore about 600 km2. However, to be consistent with Bony et al. (2020) as requested by Reviewer #2, the scale factor has been adapted. This is now explicitly mentioned in the caption of Fig. 6.

- ✅ [p 9 l 150] "we applied a threshold of 0.1 on the frequencies":

  ○ ✅ This is a bit unclear. Above you talk about the "percentage of agreement" but here of "frequencies of the level 3 dataset". How do you go from "percentage of agreement" to "frequency"? Are they the same? Does a threshold of 0.1 mean that only 10% of participants needed to say that they label an area as a given pattern? Is that a reasonable number? It seems quite low to me, wouldn't that lead to very large masks? What happens if two users classify a given pixel with two different labels (I don't think

that is taken into account with the current calculation, but the word "frequency" suggests to me that it should)?

**Response**

Thank you for pointing out this ambiguity. *Percentage of agreement* and *frequencies* have been used interchangably here. To clear this ambiguity, only *percentage of agreement* is now used in the updated manuscript.

A threshold of 0.1, which indeed means that only 10% of the participants needed to classify a certain pattern, is chosen to eliminate the most extreme variations of classifications, but still show the reader the variability in classifications. A more conservative threshold of 0.5 is now shown in Fig. 7 as well and reveals more clearly the four regimes that occured during January - February 2020 by removing most of the overlap.

- ○ ✅ Also, having a cumulative area fraction larger that 1.0 is quite confusing. It would be good to discuss what that means and why it is reasonable. How would I read from Figure 7 what area fraction is unclassified?

  **Response**

  Fig. 7 is a daily snapshot and several patterns may occur throughout the course of the day (in the updated version only from 12-20 UTC though). The overlap of the classifications therefore arises naturally but can also be caused by disparities between classifications of different or even the same user. The latter can however be reduced by increasing the threshold of agreement as seen in the updated Fig. 7. In addition, the agreement on "unclassified" regions have been included.

- ✅ [p 9 l 162] "While the Iorg/S metric is computationally cheap and can be easily applied to different regions, the manual classifications are naturally more accurate":

  - ○ This doesn't quite follow for me. Figure 7 isn't attempting to produce classification into the four organisation patterns using only Iorg/S, and so I don't think this analysis shows that it isn't possible with just Iorg/S.

  - ○ **Response**

    The paragraph has been rewritten to:

    > Overall, the different classification methods agree well with each other and no large discrepancies are found. This reassures that these methods are valid for further analysis of meso-scale patterns. While the Iorg/S metric is computationally cheap and can be easily applied to different regions, it is less suitable for short time periods because the terciles that attribute the Iorg/S pairs to specific quadrants are not robust for a small sample size. Tercile bounds from other studies can only be used if the resolution of the dataset is the same because Iorg and S depend on the resolution. For short timeperiods, like those of typical campaigns where people are interested on specific days or even sub-daily variations, manual classifications are advantageous as they do not require a large sample size. The neural network approach is a good possibility to extend the classifications to different time periods and regions by using manual classifications as a training set. Here, the manual classifications served as an additional independent validation dataset and proved once more the capabilities of the neural network which has been trained on the dataset of Rasp et al. (2020).

- ✅ [p 9 l 164] "manual classifications are most accurate":

  - ○ I would like to understand a little better how you draw this conclusion. What is your measure of accuracy? Aren't the manual classifications being used as "truth" here? If so, and assuming that any other method will classify differently in some way, how can any other method predict something better than what is being used as the reference?

  - ○ **Response**

The paragraph has been rewritten for clarification. See also response to previous comment.

- ✅ [p 10 | 179] "demands further investigations on how":

  - should be "demands further investions into how…"